# Epitaxial growth of metal-organic framework nanosheets into single-crystalline orthogonal arrays

Yingying Zou[1], Chao Liu[1] ✉, Chaoqi Zhang[1], Ling Yuan[1], Jiaxin Li[1], Tong Bao[1], Guangfeng Wei ●[2] ✉, Jin Zou ●[3] ✉ & Chengzhong Yu ●[1,4] ✉

Construction of two-dimensional nanosheets into three-dimensional regular structures facilitates the mass transfer and exploits the maximum potential of two-dimensional building blocks in applications such as catalysis. Here, we report the synthesis of metal-organic frameworks with an orthogonal nanosheet array. The assembly involves the epitaxial growth of single crystalline metal-organic framework nanosheets with a naturally non-preferred facet exposure as the shell on a cubic metal-organic framework as the core. The nanosheets, despite of two typical shapes and crystallographic orientations, also form a single crystalline orthogonally arrayed framework. The density and size of nanosheets in the core-shell-structured composite metal-organic frameworks can be well adjusted. Moreover, metal-organic frameworks with a single composition and hollow orthogonal nanosheet array morphology can be obtained. Benefiting from the unusual facet exposure and macroporous structure, the designed structure exhibits improved electrocatalytic oxygen evolution activity compared to conventional nanosheets.

Two-dimensional (2D) nanosheets have attracted particular attention because of their excellent properties including high surface to volume ratio, enhanced surface exposure and superior charge transfer ability[1-3]. The restacking of nanosheets is a general challenge that deteriorates their performance. Therefore, nanosheets with various compositions, e.g., zeolite[4], graphene[5], layered double hydroxide[6] and polymers[7], have been assembled to form 3D structures. As an emerging class of 2D materials, the assembly of metal-organic framework (MOF) nanosheets is also crucial[8-10]. MOFs nanosheets have been assembled into flower-like architectures[11-15], or grown on selected substrates (e.g., carbon fiber, nickel foam)[16-20]. In these reports, the MOF nanosheets are either randomly assembled without regular spatial arrangement, or densely stacked parallelly in one direction. To address this issue, an orthogonal nanosheet array stands out[21], where

the nanosheets are normally connected to each other with rigid angles (e.g., 90°). Such an architecture is expected to enable the full applications of 2D nanosheets by avoiding restacking and to provide large space for active site exposure and mass transfer. However, it is only reported in few examples of self-pillared 2D zeolites assembled via repetitive branching[22-25].

Herein, we report the orthogonal assembly of MOF nanosheets into a single-crystalline array via epitaxial growth (Supplementary Fig. 1). The synthesis of orthogonal nanosheets with single-crystalline arrays (ONSA) involves the oriented growth of two types of CuFe PBA (prussian blue analogs, space group of Fm3̄m, $a = 10.10$ Å, $Cu_3[Fe(CN)_6]_2$) nanosheets as the shell on the surface of cubic HKUST-1 (Hong Kong University of Science and Technology, space group of Fm3̄m, $a = 26.34$ Å, $C_{18}H_{12}Cu_3O_{15}$) as the core. The epitaxial

[1]School of Chemistry and Molecular Engineering, East China Normal University, Shanghai 200241, PR China. [2]Shanghai Key Laboratory of Chemical Assessment and Sustainability, School of Chemical Science and Engineering, Tongji University, Shanghai 200092, PR China. [3]Materials Engineering and Centre for Microscopy and Microanalysis, The University of Queensland, Brisbane, QLD 4072, Australia. [4]Australian Institute for Bioengineering and Nanotechnology, The University of Queensland, Brisbane, QLD 4072, Australia. ✉e-mail: cliu@chem.ecnu.edu.cn; weigf@tongji.edu.cn; j.zou@uq.edu.au; czyu@chem.ecnu.edu.cn

relationship between CuFe PBA and HKUST-1 results in a single-crystalline orthogonally arrayed framework of CuFe PBA nanosheets with controllable nanosheet densities and sizes. Different from reported CuFe PBA nanosheets with (001) as the most exposed surfaces[26,27], the obtained nanosheets have a naturally non-preferred (111) dominant facet exposure. By further selective etching treatment, single CuFe PBA hollow ONSA (H-ONSA) with a self-standing hollow structure can be obtained. Theoretical calculation indicates that compared to CuFe PBA nanosheets with (100) exposed facets, those with (111) exposed facets offer abundant unsaturated binding sites and stabilize the intermediate species via hydrogen bonding in oxygen evolution reaction (OER). Together with the promoted active site exposure and mass transfer by macroporous array of nanosheets, the OER performance of H-ONSA is enhanced than conventional CuFe PBA nanosheets.

## Results

### Structural characterization of ONSA-HC-1

HKUST-1 microcubes with exclusively {001} exposed facets were prepared by a solvothermal method[28,29]. Scanning electron microscopy (SEM) study shows that HKUST-1 possesses a smooth cubic morphology with sizes ranging from 1.7 to 3.0 μm (Supplementary Fig. 2a). Transmission electron microscopy (TEM) investigation indicates its solid nature (Supplementary Fig. 2b). In the X-ray diffraction (XRD) pattern, the diffraction peaks match well with the simulated ones, confirming the formation of crystalline HKUST-1 (Supplementary Fig. 2c).

Subsequently, the HKUST-1 microcubes were immersed in a mixed solution of $H_2O$ (3 mL) and ethanol (12.5 mL) for pre-activation[30]. Negligible morphology and structural changes were evidenced by SEM, TEM, and XRD characterizations (Supplementary

Fig. 3). Further addition of $K_3Fe(CN)_6$ aqueous solution triggers the growth of shell CuFe PBA nanosheets on the {001} surface of HKUST-1, forming one HKUST-1@CuFe PBA ONSA heterostructure (ONSA-HS-1). As shown in Fig. 1a, ONSA-HS-1 presents a uniform cubic morphology with porous surface. The particle size range of ONSA-HS-1 is from 3 to 6 μm, larger than that of HKUST-1. At higher magnification (Fig. 1b,c and Supplementary Fig. 4), two types of semi-hexagonal nanosheets can be seen on the cube surface with an average thickness of ≈50 nm. Type I (marked with red line in Supplementary Fig. 4b,d) is dominant and exposes three sides, and Type II (marked with blue line) is minor with four sides exposed. These nanosheets are either parallel or perpendicular to the edges of HKUST-1 cube, showing a rigid ONSA architecture. Figure 1d is a TEM image of ONSA-HS-1 particles, and confirms the core-shell heterostructure with nanosheets perpendicularly adhered on a solid HKUST-1 cube. The orientation of nanosheets is in accordance with the SEM investigation. Figure 1e is an enlarged TEM image, in which the height of outer nanosheets is estimated to be ≈0.4 μm. Similar to the SEM results, semi-hexagonal nanosheets with two different shapes (type I and II) are also observed.

To understand the chemical characteristics of obtained ONSA-HS-1, high-angle annular dark field scanning TEM (HAADF-STEM) was performed. Figure 1f shows a HAADF-STEM image of a typical ONSA-HS-1 and corresponding element maps, in which the Cu is distributed over the entire particle, while the Fe predominately exists in the outer shell region, indicating that the nanosheets in the shell have a Cu and Fe rich composition (Supplementary Fig. 5a,b) with a Cu/Fe molar ratio of ≈3.3:2 (Supplementary Fig. 5c), close to the theoretical value (3:2) of CuFe PBA ($Cu_3[Fe(CN)_6]_2$)[31]. The XRD investigation of ONSA-HS-1 was also performed, and the result is shown in Fig. 1g. As can be seen, two sets of diffraction peaks corresponding to HKUST-1 and CuFe PBA

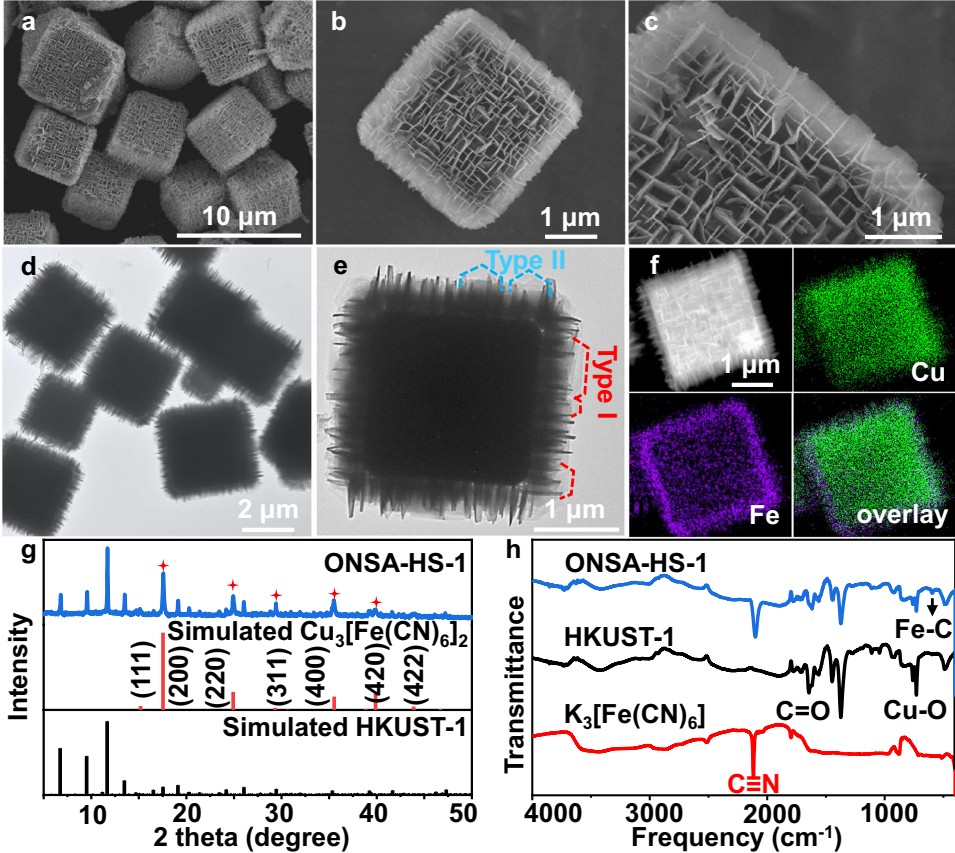

**Fig. 1 | Structural characterization of ONSA-HC-1. a–c** SEM, **d, e** TEM, **f** HAADF-STEM and corresponding element mapping images, and **g** XRD pattern of ONSA-HS-1. **h** FTIR spectra of $K_3[Fe(CN)_6]$, HKUST-1 and ONSA-HS-1.

(PDF#86-0513) can be indexed. Moreover, the Fourier transform infrared (FTIR) spectrum of ONSA-HS-1 (Fig. 1h) showed characteristic bands usually observed in HKUST-1 (e.g., carboxy group at 1645 $cm^{-1}$ and Cu-O vibration at 729 $cm^{-1}$)[32] and CuFe PBA (e.g., C≡N stretching mode at 2117 $cm^{-1}$)[33]. In addition, the inductively coupled plasma-optical emission spectrometry (ICP-OES) shows a Cu/Fe molar ratio of 5.5:1.0. According to the molecular formulas of CuFe PBA ($Cu_3[Fe(CN)_6]_2$) and HKUST-1 ($C_{18}H_{12}Cu_3O_{15}$), the mass ratio of CuFe PBA and HKUST-1 in ONSA-HS-1 is calculated to be ≈2.7:7.3.

To study the porous structure, ONSA-HS-1 was characterized by nitrogen sorption analysis in comparison with individual HKUST-1 and CuFe PBA. CuFe PBA nanocrystals with an irregular morphology were synthesized by using $Cu(NO_3)_2$·$3H_2O$ as the copper source to replace HKUST-1 (Supplementary Fig. 6a–c), indicating the crucial role of HKUST-1 substrate for the formation of ONSA-HS. The $N_2$ sorption isotherms of HKUST-1 (Supplementary Fig. 7a) show a type I hysteresis loop with a sharp condensation step at low relative pressure ($P/P_0 < 0.05$), indicating the micropore-dominant nature. By incorporating nonporous CuFe PBA, the $N_2$ uptake at low relative pressure ($P/P_0 < 0.05$) for ONSA-HS-1 is decreased. The $N_2$ sorption isotherms of ONSA-HS-1 also exhibit a capillary condensation step at high $P/P_0$ range (>0.9) with a hysteresis loop, indicative of large pores generated by the orthogonal arrangement of CuFe PBA nanosheets[34]. Then, according to the Brunauer–Emmett–Teller specific surface area of bare HKUST-1 (1037 $m^2 g^{-1}$) and CuFe PBA (53 $m^2 g^{-1}$), the specific surface area (870 $m^2 g^{-1}$) of HoC-HS-1 is close to the calculated value (771 $m^2 g^{-1}$) based on mass ratio of HKUST-1 and CuFe PBA in HoC-HS-1. The additional specific surface area in our experimental value may be attributed to the formation of CuFe PBA nanosheets and the orthogonal arrangement. The total pore volume of ONSA-HS-1 (0.77 $cm^3 g^{-1}$) is higher than that of HKUST-1 (0.57 $cm^3 g^{-1}$) and CuFe PBA (0.07 $cm^3 g^{-1}$, Supplementary Fig. 7b and Table 1), also suggesting the formation of additional pores. The average diameter of the large pores is determined to be ≈100 nm from the pore-size distribution curves in Supplementary Fig. 7b, slightly smaller than the value (≈125 nm) measured by the SEM images (Supplementary Fig. 8). Collectively, the above results indicate that ONSA-HS-1 has a core−shell heterostructure: the core is the HKUST-1 cube and the is composed of CuFe PBA nanosheets arranged in an ONSA architecture.

## Formation mechanism of ONSA-HC−1

To explore the nucleation of CuFe PBA on HKUST−1, the sample after 10 min of CuFe PBA growth was carefully selected and characterized. As shown in Supplementary Figs. 9a, b and 10, high-density CuFe PBA nanosheets with a Cu/Fe ratio of 3:2 are distributed on the surface of HKUST−1. The enlarged SEM and TEM images (Supplementary Fig. 9c, d) show two types of semi-hexagonal nanosheets. The height and thickness are estimated to be ≈180 and ≈20 nm, respectively, smaller than those of ONSA-HS-1. These observations suggest that the nucleation of orthogonal nanosheet of ONSA-HS-1 has already been imitated at this stage. The further growth of nanosheets mainly leads to their size expansion with unchanged shape and composition.

It is noted that there are two types of morphologies of nanosheets as marked in Fig. 1e. To understand their crystallographic characteristics, individual CuFe PBA nanosheets were peeled from ONSA-HS-1 particles by ultrasonication treatment, then investigated by TEM. Figure 2a, c is representative TEM images of two types of nanosheets after extensive TEM search (Supplementary Fig. 11), both with semi-hexagonal shapes. Their corresponding selected area electron diffraction (SAED) patterns are shown in Fig. 2b,d. Interestingly, both SAED patterns are typical single-crystal ED patterns that can be indexed as [1$\bar{1}\bar{1}$] zone-axis of the CuFe PBA phase, suggesting the extensive surfaces of nanosheets are {111} facets. By corelating the TEM images and corresponding ED patterns, the flat edges of the nanosheets can be indexed to {110} planes of the CuFe PBA phase in both cases. The innermost diffraction spots are indexed as 1/3{220} in both cases, similar to the observation in the case of $Ni_2Cr$[35].

To further clarify the crystallographic orientation of nanosheets in ONSA-HS-1, the CuFe PBA nanosheets area of ONSA-HS-1 was selected for ED (marked by yellow circle in Fig. 2e). The obtained diffraction spots can be indexed as two superimposed [1$\bar{1}\bar{1}$] zone-axis single-crystal ED patterns with a relevant 30° rotation (Fig. 2f), which is consistent with the superposition of diffraction spots of Type I and Type II nanosheets. It is noted that the diffraction spots associated with Type I have stronger intensity than that with Type II. Moreover, this observation is consistent in the SAED patterns from nanosheet locations of other ONSA-HS-1, as shown in Supplementary Fig. 12. By quantifying the spot intensity using a software ImageJ, the SAED spot intensity ratio of Type I versus Type II was calculated to be 9.92 ± 1.43 (Supplementary Fig. 13). Considering both Type I and II nanosheets

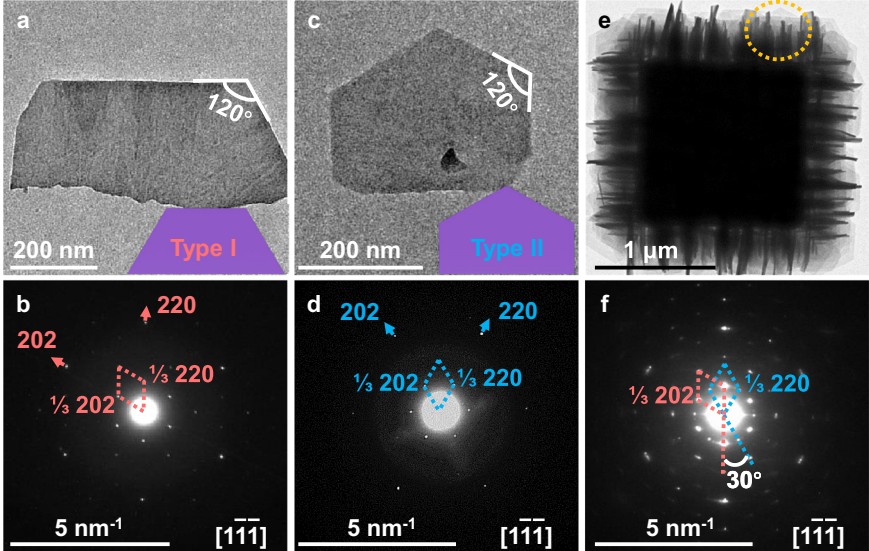

**Fig. 2 | Growth orientation of nanosheets in ONSA-HS-1. a, c, e** TEM images and corresponding. **b, d, f** SAED patterns of detached CuFe PBA nanosheet with Type I structure (**a, b**). Type II structure (**c, d**) and CuFe PBA nanosheets area in ONSA-Hs-1 (**e, f**), respectively. The inset purple semi-hexagons in (**a**) and (**c**) represent the CuFe PBA nanosheet with Type I and II structure, respectively.

have comparable sizes, their ED spot intensity ratio can be used as a rough estimation of their content difference. Therefore, Type I nanosheets are dominant in ONSA-HS-1 over Type II, in accordance with both SEM and TEM observations.

It is inferred that for all nanosheets in ONSA-HS-1, regardless of Type I or II, their {111} surfaces are either perpendicular or parallel to the electron beam with other orientation negligible. For the nanosheets parallel to the electron beam, they may be too thick for electrons to penetrate to generate electron diffraction. Nevertheless, compared to the SAED patterns of individual CuFe PBA nanosheets, the diffraction spots in ONSA-HS-1 are slightly elongated, suggesting slight misorientation of the nanosheets.

To understand the observed epitaxial relationship of both types of CuFe PBA nanosheets on the {001} facets of cubic HKUST−1, we consider the structural models of CuFe PBA and HKUST−1. Figure 3b shows a projected (110) plane of CuFe PBA, in which the vertical relationships between $(1\bar{1}\bar{1})$, $(1\bar{1}2)$ and (110) crystal planes can be seen. The exposed surfaces of CuFe PBA nanosheets can be indexed to {110} and {$1\bar{1}\bar{1}$} planes in both types[36], as shown in Fig. 3c, d. Crystallographically, the contact interface with HKUST−1 is a {110} facet for Type I and {$1\bar{1}2$} planes for Type II CuFe PBA nanosheets. Consequently, the epitaxy relationship between Type I CuFe PBA and HKUST-1 can be established as {112}$_{CuFe\ PBA}$∥{010}$_{HKUST−1}$ and {111}$_{CuFe\ PBA}$∥{100}$_{HKUST-1}$ (Fig. 2e), while that between Type II CuFe PBA and HKUST-1 as {110}$_{CuFe\ PBA}$∥{010}$_{HKUST-1}$ and {111}$_{CuFe\ PBA}$∥{100}$_{HKUST−1}$ (Fig. 2f). The theoretical lattice mismatches between $6 \times d${112}$_{CuFe\ PBA}$ and $d${100}$_{HKUST-1}$ and between $5 \times d${111}$_{CuFe\ PBA}$ and $d${010}$_{HKUST−1}$ are 6.16% and 10.66% in Type I (Fig. 2g), respectively. For Type II nanosheets, the theoretical lattice mismatch between $4 \times d${110}$_{CuFe\ PBA}$ and $d${100}$_{HKUST−1}$ is 8.42%, slightly larger than that between $6 \times d${112}$_{CuFe\ PBA}$ and $d${100}$_{HKUST-1}$ in Type I. Furthermore, each HKUST-1 microcube has six identical {001} facets, allowing the growth of CuFe PBA nanosheets at two orthogonal directions, i.e., either parallel or perpendicular to the edges of {001} surfaces of HKUST-1. This growth pattern is also independent of the type of CuFe PBA nanosheets, leading to the construction of ONSA morphology.

In reported CuFe PBA materials[26,27], (001) facets are the most exposed surfaces, but nanosheets with (111) dominant facet exposure are rare. To understand the underlying reason, the surface energies of CuFe PBA (111) and (001) surfaces were evaluated by density functional theory (DFT) calculations. The results show that the surface energy of

(001) surface (0.19 J m$^{-2}$) is 0.17 J m$^{-2}$ lower than that of (111) facet (0.36 J m$^{-2}$). This explains why nanosheets with (001) dominate facet exposure were formed during solution synthesis. In our work, however, the formation of CuFe PBA nanosheets on HKUST-1 surface is restricted by the epitaxial relationship rather than determined by thermodynamics, thus CuFe PBA nanosheets with unusual (111) dominant facet exposure are obtained (Fig. 3).

Except for the directional growth, the formation of the semihexagonal nanosheet morphology of CuFe PBA is also deserved an investigation. First, the relatively large lattice mismatch between the $(1\bar{1}\bar{1})$ plane and HKUST-1 restricts the growth along both $\pm[1\bar{1}\bar{1}]$ directions, and the lattice mismatch along the directions perpendicular to the $[1\bar{1}\bar{1}]$ direction is less significant in Type I than Type II nanosheets. Thus, (111) exposed CuFe PBA exhibit a nanosheet morphology and more Type I nanosheets should be observed in ONSA-HS-1. According to the Wulff's theory[37], for a crystal grown on a substrate, the equilibrium shape of the crystal is determined by minimization of the surface and interfacial energy. The Wulff's point is inside and outside the substrate (HKUST-1) for Type I and II nanosheets observed in Fig. 2b,e, respectively. This also suggests a relatively higher interfacial energy in Type II than I, thus a higher portion of exposed {110} facets is observed to minimize the interfacial energy.

## Synthesis of MOFs ONSA with tunable structures

The structure of ONSA-HS can be further tuned by altering the pre-activation strength of HKUST-1 using different $H_2O$ amounts in the first step. By decreasing the $H_2O$ amount from 3 to 1 mL, the resultant product (denoted as ONSA-HS-2) possesses a uniform ONSA architecture with decreased nanosheet density and increased height (≈445 nm, Fig. 4a, c, Supplementary Figs. 14a and 15a) compared to ONSA-HS-1. For the synthesis without $H_2O$ pre-activation (the sample was denoted as ONSA-HS-3), the orthogonal nanosheets with further increased height (650 nm) are vertically grown on the surface of HKUST-1 with a sparser arrangement (Fig. 4I–k, Supplementary Figs. 14b and 15b). This suggests that decreasing the $H_2O$ amount restricts the nucleation of the CuFe PBA nanosheets, leading to a lower number of nucleates and the grown nanosheets with larger sizes. The XRD patterns (Supplementary Fig. 16) and EDX element analysis (Fig. 4d–h, l–p, and Supplementary Fig. 17a,b) of ONSA-HS-2 and 3 further indicate the successful formation of HKUST-1@CuFe PBA heterostructures.

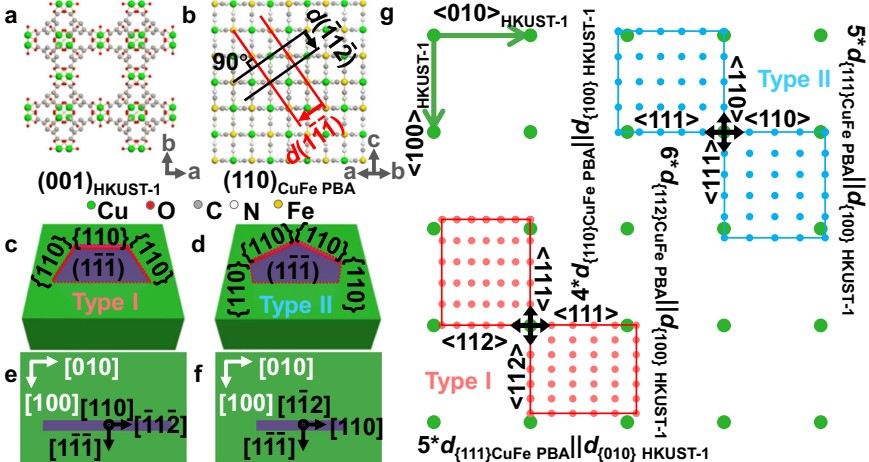

**Fig. 3 | Epitaxial relationship between CuFe PBA and HKUST-1. a** (110) plane and **b** (001) plane of HKUST-1. The green, red, gray, white and yellow balls in (**a**) and (**b**) represent Cu, O, C, N and Fe atoms, respectively. The red lines and black lines in (**b**) represent the $(1\bar{1}\bar{1})$ and $(1\bar{1}2)$ planes of CuFe PBA, respectively. **c** Scheme of the type I grown on (001) plane of HKUST-1. **d** Scheme of the type II grown on (001) plane of HKUST-1. **e** Top view of crystal structure showing (110) of CuFe PBA. **f** Top view of crystal structure showing $(1\bar{1}2)$ of CuFe PBA. HKUST-1 and CuFe PBA are marked by green and red color, respectively. **g** Illustration of the epitaxy relationship between CuFe PBA and HKUST-1. The green, red, and blue dots in (**g**) represent the lattice points of (001) plane of HKUST-1, (110) and $(1\bar{1}2)$ planes of CuFe PBA, respectively.

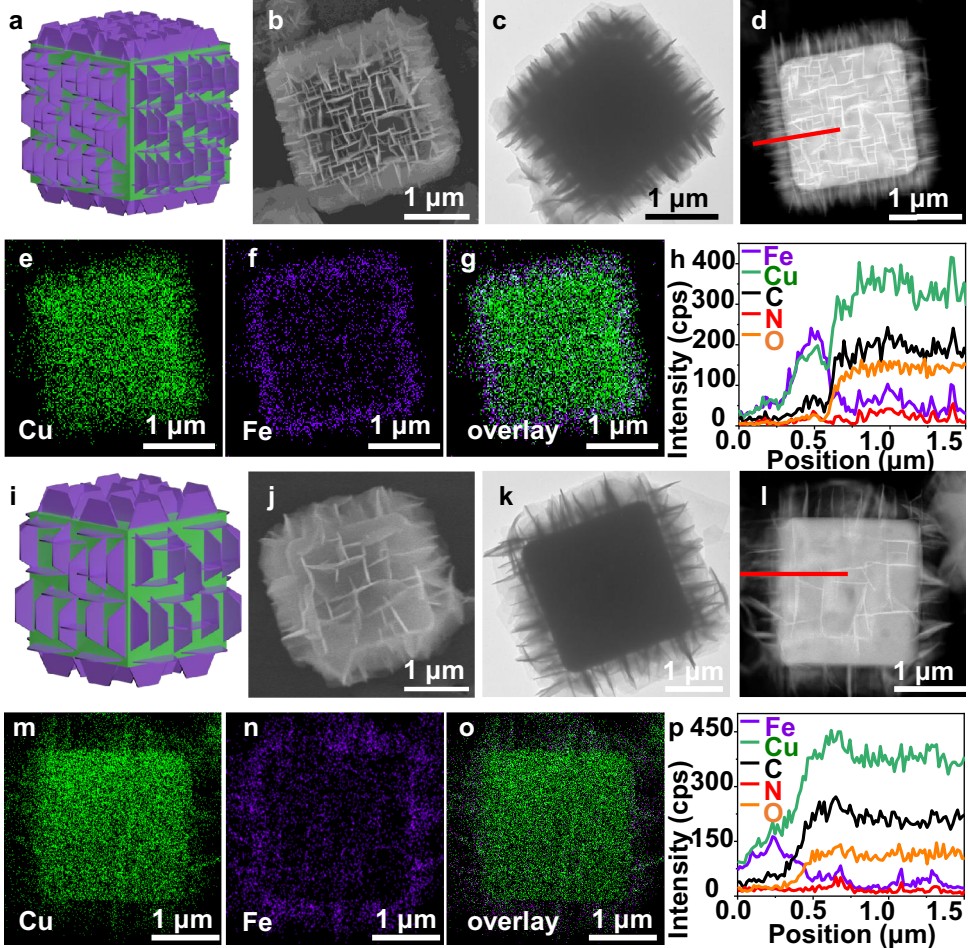

**Fig. 4 | Controllable synthesis of ONSA-HS by regulating the activation degree.** **a, i** Structural illustration. **b, j** SEM. **c, k** TEM. **d, l** STEM and **e–g, m–o** corresponding elements mapping images and **h, p** line scan patterns along the red line in (**d**) and (**l**) of ONSA-HS-2 and 3. HKUST-1 and CuFe PBA are marked by green and red color in (**a**) and (**i**), respectively.

To gain insight into the role of pre-activation, HKUST-1 samples after pre-activation treatment were collected and characterized by X-ray photoelectron spectroscopy (XPS). As shown in Supplementary Fig. 18a, the Cu 2$p$ spectrum of pristine HKUST-1 without pre-activation can be fitted into Cu 2$p_{1/2}$ and Cu 2$p_{3/2}$ orbitals of Cu$^+$, Cu$^{2+}$, and satellite peaks. According to ref. 38, the existence of Cu$^+$ species should be attributed to the formation of ligand insufficient defective Cu sites with lower coordination (<4) and reduced total charge. The molar ratio of Cu$^+$/Cu$^{2+}$ can be estimated as 29% from Supplementary Fig. 18a. By increasing the H$_2$O amount in the pre-activation treatment to 1 and 3 mL, the Cu$^+$/Cu$^{2+}$ ratio in activated HKUST-1 increases to 32% and 40%, respectively (Supplementary Fig. 18b–d), indicating more defective Cu sites are generated, possibly due to the replacement of coordinated ligands by H$_2$O[28]. Meanwhile, higher concentration of released Cu ions was detected in the supernatants with more added H$_2$O (Supplementary Fig. 19).

The coordination unsaturated Cu sites with higher activity are anticipated to act as the nucleation sites to bond with [Fe(CN)$_6$]$^{3-}$ for inducing the growth of CuFe PBA. The similar role of Cu atoms on the {100} facets of HKUST-1 has also been reported for the interfacial growth of MOF-5 via forming Cu$_{(HKUST-1)}$-O$_{(MOF-5)}$ bonds[39]. Therefore, strong pre-activation results in increased nucleation sites and Cu$^{2+}$ sources, contributing to the growth of CuFe PBA nanosheets with higher density and smaller sizes. The lower density of CuFe PBA nanosheets (Supplementary Fig. 20a-h) at the initial stage with 10 min reaction time during synthesis of ONSA-HS-2 and 3 than ONSA-HS-1

further support the pre-activation strength dependent growth of CuFe PBA. By further increasing the H$_2$O amount to 5 mL, an inhomogeneous mixture (denoted as ONSA-HS-4) composed of individual nanosheets is obtained (Supplementary Fig. 21), indicating that homogeneous ONSA-HS can be synthesized only in a certain range of H$_2$O used in the pre-activation step.

Apart from the controllable synthesis of ONSA-HS, CuFe PBA ONSA with a hollow structure (H-ONSA) were obtained as a secondary product of ONSA-HS-1 by selective etching of HKUST-1 with lower stability. As shown in Fig. 5b, c, H-ONSA shows a similar ONSA architecture to ONSA-HS-1. Differently, a hollow cavity with size of 2.2 μm is generated in a typical H-ONSA (Fig. 5d). A self-standing hollow ONSA architecture composed of CuFe PBA nanosheets can be produced without the support of HKUST-1, indicating the high structural robustness of CuFe PBA ONSA architecture. The HAADF-STEM and corresponding element mapping images (Fig. 2e–h) show the homogeneous distribution of Cu, Fe, C, and N elements in the framework of H-ONSA, which is further evidenced by the EDX line scanning results (Fig. 2i). The XRD pattern of H-ONSA (Fig. 5j) shows only one group of diffraction peaks assigned to CuFe PBA, verifying the removal of HKUST-1.

The reports on interfacial growth of MOF nanosheets can be approximately divided into two categories: (1) non-epitaxial growth of MOF nanosheets without regular spatial arrangement[16,19,20,40]; (2) epitaxial growth of densely stacked MOF nanosheets along one direction on a low-symmetrical MOF substrate[41]. The key difference in our

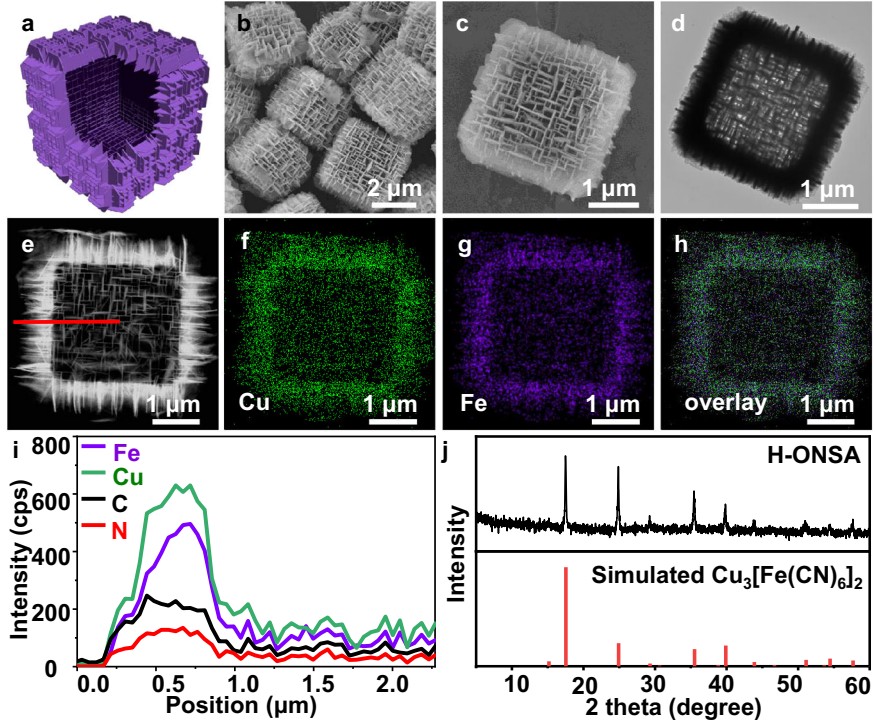

**Fig. 5 | Structural characterization of H-ONSA. a** Structural illustration, **b, c** SEM, **d** TEM, **e** STEM and **f–h** corresponding elements mapping images, **i** line scan patterns collected along the red line in (**e**) and (**j**) XRD pattern of H-ONSA.

design is the utilization of a high-symmetrical MOF (HKUST-1 cubes with a space group of Fm3̄m) as the substrate with six identical exposed {001} facets. One each facet, the epitaxial growth of CuFe PBA nanosheets occurs at two orthogonal directions, either parallel or perpendicular to the edges of {001} surfaces. This allows orthogonally assembled MOF nanosheets into a single-crystalline array with large pores in the 3D framework, which is rarely reported in MOF-based nanosheet assemblies. Moreover, the MOF-based ONSA is also distinctive from self-pillared zeolite orthogonal nanosheet array formed by self-branching, usually without a uniform morphology[22–25].

## Electrocatalytic performance

OER plays important roles in various oxygen-involved energy conversion and storage technologies, where the development of effective electrocatalysts is the key[42,43]. Herein, the OER performances of H-ONSA were evaluated in 1.0 M KOH electrolyte with CuFe PBA nanocrystals, peeled CuFe PBA nanosheets (denoted as nanosheets-111, Supplementary Fig. 22) and commercial $RuO_2$ catalyst. In addition, to explore the effect of facet exposure, (001) dominated CuFe PBA nanosheets (denoted as nanosheets-001, Supplementary Figs. 23 and 24) were also synthesized. Figure 6b shows the linear sweep voltammetry (LSV) curves of all samples. The H-ONSA requires lowest overpotential of 241 mV to reach the current density of 10 mA cm$^{-2}$, superior to those of CuFe PBA nanocrystals (430 mV), nanosheets-001 (350 mV), nanosheets-111 (301 mV), and even commercial $RuO_2$ (258 mV), indicating the highest OER activity (Fig. 6c). In addition, the OER kinetics of four catalysts were examined by Tafel plots (Fig. 6d). The H-ONSA exhibits a lower Tafel slope of 52.1 mV dec$^{-1}$ than nanocrystals (115.7 mV dec$^{-1}$), nanosheets-001 (96.0 mV dec$^{-1}$), nanosheets-111 (94.3 mV dec$^{-1}$) and $RuO_2$ (67.5 mV dec$^{-1}$), showing accelerated OER kinetics. To assess the intrinsic activity of H-ONSA, the turnover frequency (TOF, see the calculation details in Methods) including lower-bound TOF (TOF$_{lb}$) and upper-bound TOF (TOF$_{ub}$) was calculated to be $1.84 \times 10^{-4}$ and 352.59 s$^{-1}$ at an overpotential of 240 mV, respectively. The OER performance of H-ONSA is comparable to most reported PBA-

derived electrocatalysts or other MOFs and transition metal-based materials (Supplementary Table 2).

Apart from the activity, stability is also crucial for OER electrocatalysts. Successive CV with a scan rate of 50 mV s$^{-1}$ was carried out. After 500 cycles, the obtained LSV curve of H-ONSA is similar to the initial one with slight increase of the overpotential (Fig. 6g), displaying a reasonable stability. The long-term durability of H-ONSA was further assessed by the chronoamperometric measurements (Fig. 6h). After 24 h, only about 7.4% of current decay is observed in H-ONSA, significantly smaller than that of $RuO_2$ (high up to 44.5%), suggesting the excellent OER stability. The sample after cycling test was also characterized by TEM, SEM and XRD, showing well-preserved morphology and crystalline structure (Supplementary Fig. 25).

The changes of the surface structure and chemical states of H-ONSA before and after OER test were further studied by XPS (the sample after use was denoted as H-ONSA-A). In the Fe 2p spectrum of H-ONSA (Supplementary Fig. 26a), the peaks at 708.2 and 721.0 eV are assigned to $Fe^{2+}$. In addition, two peaks at 710.0 and 723.7 eV are attributed to the $2p_{3/2}$ and $2p_{1/2}$ states of $Fe^{3+}$, with two satellite peaks at 787.9 and 804.1 eV. The $Fe^{3+}/Fe^{2+}$ ratio was determined to be 1.11. The Cu 2p spectrum of H-ONSA (Supplementary Fig. 26b) shows four main peaks, attributed to $2p_{1/2}$ and $2p_{3/2}$ orbitals of $Cu^{2+}$ (955.8 and 935.8 eV) and $Cu^+$ (952.7 and 933.0 eV), and satellite peaks with $Cu^{2+}/Cu^+$ ratio of 0.71. Compared to the fresh H-ONSA, H-ONSA-A exhibited similar Fe and Cu species with higher $Fe^{3+}/Fe^{2+}$ and $Cu^{2+}/Cu^+$ ratios (1.44 and 2.03, respectively). For O 1s spectrum of H-ONSA (Supplementary Fig. 26c), the peak at 532.5 eV is assigned to the -OH group. The observation of a new peak of -OOH group at 535.4 eV for H-ONSA-A indicates the generation of metal (oxy)hydroxides during OER process, consistent with the literature reports of MOF-based OER electrocatalysts[44,45].

To understand the superior OER performance of H-ONSA, the electrochemical surface area (ECSA) is further studied by calculating double-layer capacitances ($C_{dl}$)[46]. The CV curves of different samples at different scan rates are presented in Supplementary Fig. 27. As expected, the H-ONSA affords more available active sites with a $C_{dl}$

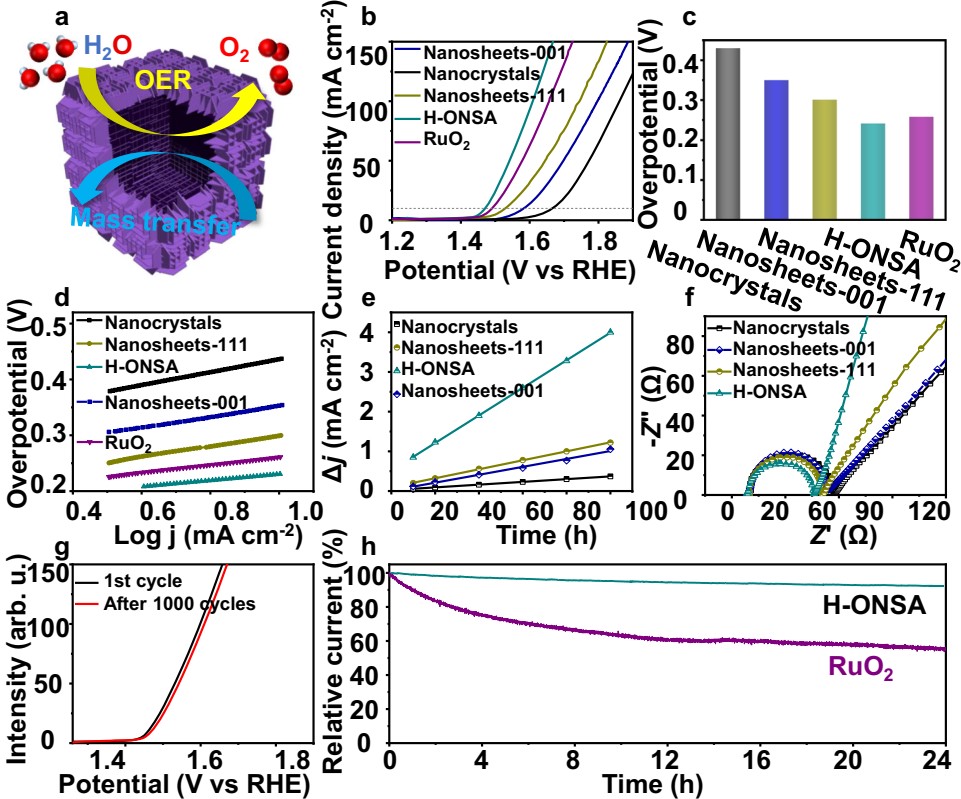

**Fig. 6 | OER performance of H-ONSA. a** Structural advantages of H-ONSA for OER; **b** LSV curves, **c** overpotentials required for current density of 10 mA cm$^{-2}$, **d** Tafel slopes from LSV curves; **e** CV current density versus scan rate of different catalysts; the linear slope is equivalent to the double-layer capacitance (Cdl); **f** EIS spectra recorded at 1.479 V versus RHE; **g** LSV curves before and after successive CV scanning; **h** long-term stability of H-ONSA and RuO$_2$.

value of 34.8 mF cm$^{-2}$ (Fig. 6e), which is ≈9.9, 3.5, and 3.1 times higher than that of nanocrystal (3.5 mF cm$^{-2}$), nanosheets-001 surface (9.97 mF cm$^{-2}$) and nanosheets-111 (11.3 mF cm$^{-2}$). Moreover, electrochemical impedance spectroscopy (EIS) was used to investigate the charge transport kinetics. The EIS spectra (Fig. 6f) show the lowest charge transfer resistance ($R_{ct}$) of H-ONSA among all samples, revealing the facilitated charge transfer.

## DFT calculations

To gain further insight into the facet exposure and OER activity relationship, DFT calculations were carried out to simulate the OER processes over (111) and (001) surfaces of CuFe PBA. The optimized structures of the OER intermediates on (111) and (001) surfaces are shown in Fig. 7a,b, respectively. The corresponding free energy reaction profiles at $U = 1.23$ V vs. SHE is shown in Fig. 7c. In line with the experimental observation that CuFe PBA nanosheets with a (111) dominant surface exhibit enhanced OER performance, the accumulated free energy cost of OOH$_{ad}$ generation, a rate-determining step during OER process[47,48], on the (111) surface is 1.10 eV, which is 0.26 eV lower than that on the (001) surface. The calculation results indicate that the superior OER activity of (111) surface can be attributed to two aspects. Firstly, the Fe atoms on the (111) surface are unsaturated and five-coordinated (as indicated by the red arrows in Fig. 7a), whereas on the (001) surface, they are saturated and six-coordinated (red arrows in Fig. 7b). As a result, the water molecules can directly adsorb onto the Fe sites on the (111) surface, while the replacement of -CN group is necessary for the water adsorption on the (001) surface. Secondly, on the (111) surface, the nearby -CN groups can stabilize the hydrogen-containing intermediate species adsorbed on the surface via the formation of hydrogen bonds. Particularly, the formation of a strong hydrogen bond (with a bond length of 1.39 Å) leads to a reduced energy cost of the rate-determining *O →*OOH step on the (111) surface

(0.40 eV), which is 0.15 eV lower than the energy cost on (100) surface (0.55 eV) in the absence of hydrogen bond formation.

## Discussion

Based on the above results, the enhanced OER performance of H-ONSA should be ascribed to the following structural superiorities (Fig. 6a). (1). The (111) dominated nanosheets offer unsaturated active sites and stabilize hydrogen-containing intermediate species, favoring the OER process. (2). The orthogonal arrangement of free-standing nanosheets reduces the likelihood of restacking of nanosheets, providing abundant surface-active sites and thus improving the catalytic activity. (3). The created vertical and penetrating pore channels[49,50], acting like a "highway", facilitates the diffusion of electrolytes and O$_2$ molecules. (4). The hollow structure is also conducive for the active site utilization and mass transfer[51,52], enhancing the electrocatalytic properties.

In conclusion, we have demonstrated the synthesis of CuFe PBA with unusual facet exposure and a single crystalline orthogonally arrayed framework, either in core-shell or hollow architectures. The structure-OER function relationship is also revealed. Our findings are expected to help designing MOF nanosheet-based architectures/heterostructures with enhanced properties.

## Methods

### Reagents and materials

Copper nitrate hexahydrate (Cu(NO$_3$)$_2$·3H$_2$O, 99%, Sinopharm Chemical Regent Co., Ltd), sodium borohydride (NaBH$_4$, 98%, Sinopharm Chemical Regent Co., Ltd), trimesic acid (C$_9$H$_6$O$_6$, 99%, Adamas-beta), lauric acid (C$_{12}$H$_{24}$O$_2$, 99%, Adamas-beta), 1-butanol (C$_4$H$_{10}$O, 99.9%, Adamas-beta), potassium hexacyanoferrate (III) (K$_3$[Fe(CN)$_6$], 99%, Adamas-beta) and ethanol (C$_2$H$_5$OH, 99.5%, Adamas-beta) were used as received. Millipore water was used in all experiments.

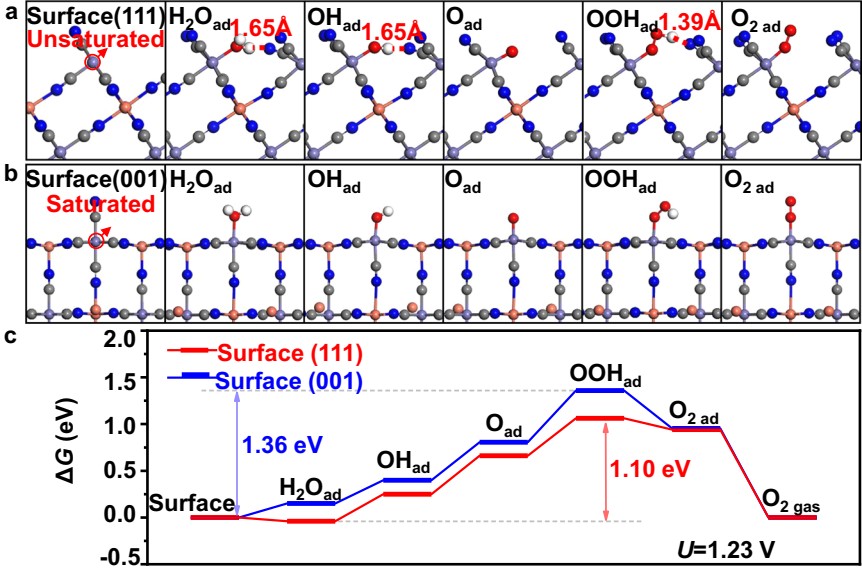

**Fig. 7 | DFT calculations.** Optimized structures of the reaction intermediates on the CuFe PBA **a** (111) surface and **b** (001) surface. **c** Free energy reaction profiles of the OER process at 1.23 V vs SHE. The purple, pink, gray, blue and white dots represent the Fe, Cu, C, N, and H atoms, respectively.

## Synthesis of HKUST-1 microcubes

The HKUST-1 microcubes were synthesized according to a previous report[28]. Typically, 123 mg (0.51 mmol) of $Cu(NO_3)_2 \cdot 3H_2O$, 72 mg (0.34 mmol) of trimesic acid ($C_9H_6O_6$) and 12.75 g (63.65 mmol) of lauric acid were dissolved in 45 mL of 1-butanol by vigorous stirring. Then, the mixture was transferred into a 100 mL Teflon-lined stainless-steel autoclave and heated at 120 °C for 5 h. Afterwards, the blue product was washed with ethanol for three times ($8000 \times g$) and re-dispersed in 17 mL of ethanol for further use.

## Synthesis of ONSA-HS

3 mL of water was added into 12.5 mL of HKUST-1 suspension (≈22 mg of HKUST-1 microcubes) with stirring for 1 h at room temperature for pre-activation. Then, 7.5 mL of $K_3[Fe(CN)_6]$ aqueous solution (1.33 g L$^{-1}$) was rapidly injected with stirring for another 1 h. The product was then collected by centrifugation and washed with water and ethanol, denoted as ONSA-HS-1. ONSA-HS-2, 3, and 4 were synthesized by adding 1, 0, and 5 mL of $H_2O$ during the activation process, respectively, with other conditions unchanged.

## Ultrasonic treatment of ONSA-HS-1

1 mg of ONSA-HS-1 was immersed into 1 mL of ethanol. By ultrasonication treatment at 40 kHz for 2 h, individual CuFe PBA nanosheets were peeled from ONSA-HS-1 particles. The product was then collected by centrifugation and washed with water and ethanol ($8000 \times g$).

## Synthesis of CuFe PBA nanosheets-001

Typically, 32.9 mg (0.1 mmol) of $K_3Fe(CN)_6$, 0.1 g (1.7 mmol) of NaCl and 0.1 g of PVP were firstly dissolved into 5 mL of $H_2O$, followed by the addition of 25 mL of DMF. Afterward, 5 mL of aqueous solution containing 20.0 mg (0.1 mmol) of $Cu(Ac)_2 \cdot H_2O$ and 58.8 mg (0.2 mmol) of sodium citrate was added into above solution with continuously stirring in an ice bath for 12 h. By further stirring at room temperature for another 36 h, the final product was collected by centrifugation and washed with water and ethanol ($8000 \times g$).

## Characterization

TEM images were obtained using Hitachi HT7700 at 120 kV. Chemical composition analyses were carried out using a JEM-2100F (JEOL, Japan) operating at 200 kV (the camera length has been carefully calibrated with polycrystalline gold nanoparticles as reference) equipped with an X-ray energy dispersive spectrometer (EDS: X-Max 80T, Oxford, UK). SEM images were acquired by scanning electron microscope (HITACHI-S4800). XRD patterns were recorded by a Bruker D8 Advanced X-Ray Diffractometer with Cu Kα radiation ($\lambda = 0.154$ nm). $N_2$ sorption isotherms were measured using Micromeritics ASAP-2460 at −196 °C. FTIR spectra were collected on a Nicolet Fourier spectrophotometer using KBr pellets. XPS studies were carried out on a Thermo ESCALAB 250 using an Al Kα radiation and $C$ 1s (284.8 eV) as a reference to correct the binding energy. The concentrations of copper ions were calculated by Agilent 730 ICP-OES. UV−vis spectra were obtained by using a UV−vis spectrophotometer (Perkin Elmer Lambda 750).

## Electrochemical measurement

Oxygen evolution reaction test was carried out in a standard three-electrode system with H-ONSA as the working electrode, platinum wire as a counter electrode, and Ag/AgCl as a reference electrode in a CHI 760E electrochemistry workstation (ChenHua Instruments Co., China). All the potentials in the measurements were calculated relative to the reversible hydrogen electrodes (RHEs) according to Eq. (1).

$$\text{Potential} = E_{\text{Ag/AgCl}} + 0.059\text{pH} + 0.197\text{V} \qquad (1)$$

The working electrode was fabricated as follows: 5.0 mg of catalyst was dispersed in 0.95 mL of ethanol and 0.05 mL of 5.0 wt% Nafion solution, and then ultrasonicated to form a relatively homogeneous ink. Then the suspension (10 μL) was dripped onto the Ni foam (1 cm$^2$). LSV were acquired through the rotating disk electrode (RDE) technique in 0.1 M KOH at a scan rate of 10 mV s$^{-1}$. The Tafel slope was calculated according to Eq. (2) as follows,

$$\eta = a + b \log j \qquad (2)$$

where $\eta$ is the overpotential, $a$ is a constant, $b$ is the Tafel slope, and $j$ refers to the current density. The EIS measurements were performed with a frequency of 100 kHz to 0.1 Hz. To evaluate the ECSAs, the double-layer capacitance ($C_{\text{dl}}$) was measured via CVs at different scan rates. The CV curves were collected in the potential range from 1.35 to 1.45 V versus RHE at different scan rates. Continuous CV scans were performed in the range of 1.1–1.6 V versus RHE at a sweep rate of 10 mV s$^{-1}$.

## Calculation of turnover frequencies[53]

Turnover frequency (TOF) is an important quantitative metric for electrocatalysts and can be calculated by Eq. (3),

$$TOF = j \times A/(4 \times F \times n) \tag{3}$$

where $j$ is the current density normalized by ECSA values and mass loading of catalyst at the given overpotential, $A$ is the area of the electrode, F is the Faraday constant (96485 C mol$^{-1}$), and $n$ is the number of moles of active sites of the metal in the catalysts, which can be calculated by two methods:(1) Lower bound TOF (TOF$_{lb}$): The TOF was calculated by assuming that all the metal sites in H-ONSA are active to catalysis according to Eq. (4):

$$n = \% mass \text{ of metal in catalyst} \times mass \text{ of catalyst/molecular weight of metal} \tag{4}$$

(2) Upper bound TOF (TOF$_{ub}$)[54]: The TOF was calculated by assuming that only the surface metal sites in the compounds are active to catalysis by using following method[55,56]: The number of surface metal atoms in the H-ONSA was calculated by assuming that the (111) crystal face is exposed in all cases. The (111) facet in $a$ unit cell gives six Fe atoms and six Cu atoms with an area of $\sqrt{3}a^2$, where $a$ (10.10 Å) is the unit cell edge length of CuFe PBA. The density of surface metal atoms is thus calculated to be $6.79189 \times 10^{18}$ surface metal atoms per m$^2$. Further according to the mass loading for OER test (0.05 mg) and surface area (113.4 m$^2$ g$^{-1}$, Supplementary Fig. 28) of H-ONSA, the number of total surface metal atoms was determined to be $3.85201 \times 10^{16}$.

## DFT calculations

Density functional theory (DFT) calculations were performed using VASP 6.2.1 packages[57] with projected augmented wave pseudo-potentials[58,59]. The exchange-correlation energy was treated based on the generalized gradient approximation by using Perdew–Burke–Ernzerhof functional[60]. The plane-wave cutoff energy was set to 450 eV. The DFT-D3 (BJ) method of Grimme[61,62] was employed to describe long-range VDW interactions. The Monkhorst-Pack scheme with a k-point separation length of 0.05 Å$^{-1}$ was utilized for sampling the first Brillion zone[63]. To correct the zero-point energy for reaction barrier, the vibrational frequency calculations were performed via the finite-difference approach. The solution effect was considered by the implicit solvation model implemented in the VASPsol package[64,65]. The structure model of CuFe PBA was constructed based on the experimental results. All atoms are fully relaxed during the lattice optimization. For the slab models of the (111) and (001) surfaces of CuFe PBA, the bottom two layers were fixed at the bulk-truncation position while the other layers were allowed to relax. The Quasi-Newton l-BFGS method was used for geometry relaxation until the maximal force on each degree of freedom less than 0.01 eV Å$^{-1}$. To derive the free energy reaction profiles, we followed the same approach as our previous work[66]. The standard thermodynamic data[67] were utilized to obtain the temperature and pressure contributions. The free energy of gaseous O$_2$ was derived as Eq. (5)

$$G_{[O2]} = 4.92 eV + 2G_{[H2O]} - 2G_{[H2]} \tag{5}$$

by utilizing the oxygen evolution reaction equilibrium at the standard conditions.

## Reporting summary

Further information on research design is available in the Nature Portfolio Reporting Summary linked to this article.

## Data availability

The data that support the findings of this study are available from the corresponding authors upon request.

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

## Acknowledgements

We acknowledge financial support from the National Natural Science Foundation of China NSFC 21905092 (C.L.), 22075085 (C.Y.), 22173069 (G.W.); Shanghai Municipal Natural Science Foundation (21ZR1467800, G.W.), and Fundamental Research Funds for the Central Universities.

## Author contributions

Y.Z., C.L., J.Z., and C.Y. conceived the idea. Y. Z. and C.L. performed most of material synthesis and characterizations and electrochemical measurements. C.Z., L.Y., J.L., and T.B. took part in the electrochemical measurements and discussions. G.W. carried out the DFT calculations. C.Y., C.L., J.Z., and G.W. supervised the work and directed the research.

## Competing interests

The authors declare no competing interests.
