## [Peer Review File · Nature Communications]

nature portfolio

Peer Review FileReviewer comments, first round

Reviewer #1 (Remarks to the Author):

This article by Chen-Zhong Yu et al reports on the heterostructure core-shell MOFs with house of cards architecture. In this paper, the core-shell MOF was synthesized by epitaxial growth in lattice mismatching on 2D nanosheets on 3D template MOF. In addition, the size and density of 2D nanosheets in the composite MOFs were controlled by the amount of H₂O in the synthesis, and hollow structure MOFs were synthesized by etching core MOFs with relatively low stability. Hollow structure nanosheets arrayed by templating effect exhibits improved catalytic performance compared to randomly arrayed nanosheets. This study has been well carried out, and the explanation for strategy of heterostructure MOF with house of cards architecture. However, it remains the concerns on impact and novelty of this paper in this field. Previously, strategy for the synthesis of 2D hollow nanosheet from Core-shell is reported (Dalton Trans., 2021, 50, 8179-8188, Chem. Eng. J., 2019, 370, 666-676 and Chem. Eng. J., 2022, 449, 137780). In addition, some explanation and claims are not fully supported by proper data. Therefore, it seems that an emphasis different from previous papers is needed, and detailed supplementation seems to be necessary. With the current manuscript, it is difficult to recommend the publication in Nature Communications.

1. To generalize this strategy, I recommend to replace the word "house of cards architecture". "House of cards" is an idiom which has another meaning: an unstable structure. Therefore, the expression of "Rigid house of cards architecture" on line 21 of page 4 gives the contradictory impression to readers.
2. In the abstract, the expression of "guest MOF" and "host MOF" is not intuitively acceptable based on the common sense regarding to host-guest chemistry. Thus, I recommend to replace the words with "shell MOF" and "core MOF".
3. In the Figure 1G, the XRD pattern of nanosheets in HoC-HS-1 is shifted to lower angles compared with the simulated pattern. The explanation is needed.
4. The discussion on N₂ sorption isotherm is not appropriate. By considering the material ratio of HKUST-1 and CuFe PBA in HoC-HS-1, is the isotherm of HoC-HS-1 plausible? Because CuFe PBA is nonporous, HoC-HS-1 should show the lower uptake than the current result. In addition, authors claim that the large hysteresis loop in high relative pressure ranges from 0.9 to 1.0 is attributed to the macropores by the nanosheets array. First of all, this hysteresis is hard to say large, and there is no supporting evidences for the existence of macropores. In order to confirm this, I recommend to analyze the macropores generated by the arrayed nanosheets on the core MOFs by using TEM and/or SEM.
5. In the page 7, the explanation of ultrasonication treatment is insufficient. Therefore, I recommend to add more details in the supplementary information.
6. In the page 7 to 8, I recommend that the description of calibrating the camera length with gold nanoparticles as the reference is moved to the supporting information.
7. In the page 8, There is not enough information to claim that Type 1 is dominant over Type 2. It should be explained with supporting evidences.
8. Despite the crystallographic studies, the interfacial structure between nanosheets and template MOF is still unclear. Is it expected that the chemical bonding exists? If so, how can be chemical bonding formed even though these MOFs have the lattice mismatching?
9. In Figure 4, it is necessary to state the kinds of metal in the EDS mapping data (figure 4e-g, 4m-o). Also, it would be better to match the schematic image of figure 4a,i with the color of the metal used for mapping (e.g. in figure 4a, host is purple and guest is green)

10. To claim the good catalytic performance of the hollow structure, it is need to provide the result of catalytic reaction test for core-shell.

11. Please revise and update any errors and typos in the papers.

(1) In the Figure 3, (a) (110) plane and (b) {001} plane of HKUST-1 should be replaced to (110) plane of CuFe PBA and (001) plane of HKUST-1.

(2) In the Figure 4, (d, l) STEM and corresponding elements mapping (e-j, m-o) should be replaced to (d, l) STEM and corresponding elements mapping (e-g, m-o).

(3) In the page 14, the period placed between morphology and (18-21) should be removed.

(4) In the figure S5B, the unit of X axis is an error.

(5) In the reference 19, the page number of the reference paper is an error. (1-9 is changed to 4262)

(6) In the reference 29, the page number of the reference paper is an error. (1122279 is changed to 112279)

Reviewer #2 (Remarks to the Author):

The manuscript describes the growth of CuFe Prussian Blue Analogue (PBA) nanosheets on micron scale cubic crystals of the Cu MOF HKUST-1, with the source of Cu being the MOF itself. The morphology of the composite can be controlled, changing the height, density and arrangement of the PBA nanosheets, and the interior HKUST-1 MOF can be selectively etched to leave a free-standing hollow structure. The catalytic properties of the different morphologies of the CuFe PBA material – nanosheets, nanocrystals, and hollow “house of cards” cubes – are assessed and results interpreted in terms of the accessibility of the substrate for catalytic sites in the different morphologies.

The manuscript is essentially based around discussion of an interesting and visually pleasing material, with large amounts of content devoted to describing the morphology and rationalizing its formation. I found the “house-of-cards” description to be meaningless; it seems like an attempt to dress up a material for publication by comparison with a macroscale object. The assembly looks nothing like a house of cards (which has at least short-range order), it is simply a random arrangement of sheets growing off a cubic core with some orientational preference. I would suggest removing this unnecessary comparison.

I also found the description of type I and type II nanosheets difficult to interpret when it first arises in the manuscript. The image in Figure 1e is not clear at all – it is impossible to tell the difference – whereas the schematic in Figure 2 is much clearer. What is not clear is why there is a need to make this distinction. The only place where convincing evidence of two different “forms” is provided is in the characterization of individual sheets cleaved from the MOF support. Do we know if these “types” are nanosheets growing in slightly different directions from the MOFs, or simply parts of sheets broken away from the MOF in different places? Some much clearer microscopy of the conjugates would be required to directly evidence this. In any case, I see no reason to make this distinction, which is simply based upon how the nanosheets project from the MOF, other than to facilitate a more lengthy discussion of the material structure.

The various PBA morphologies are used as catalysts for the reduction of p-nitrophenol with sodium borohydride. This is a very standard reaction which can be catalyzed by a huge range of materials, to the extent that someone has even written a review on the topic (DOI: 10.1007/s40089-021-00362-w, not cited). Why do the PBA particles stand out here? What is the comparison with alternative systems in terms of catalyst loading, kinetics, etc? It is well known that greater surface area enhances catalytic activity, so the results with the different morphologies of the PBA materials are not at all surprising.

In addition, the reporting of the catalysis experiments could be much improved. The mol % of the catalyst is not reported, and it is described as being added as “an aqueous solution”. If these

materials have dissolved fully, then surely they are no longer in the desired morphology? The authors should report both the characterization of the material after catalysis, and the Cu and Fe content of the supernatants to estimate degradation. It could simply be that the hollow cube morphology, which has the highest surface area, degrades more than the other morphologies and releases catalytic metal ions or metal oxide nanoparticles into solution (iron oxide nanoparticles are known to catalyze this reaction).

Overall, while the materials are interesting to look at and characterised in reasonable detail, the manuscript does not provide any noteworthy new results. This is simply an interesting material, which is discussed in depth and used for very rudimentary catalysis work (for which significant additional work is required), and so I cannot recommend publication in Nature Communications.

Some other minor suggestions for improving the manuscript:

1. It would be useful to provide structural composition of the two materials in the introduction for the general reader.
2. The actual composition of the PBA-HKUST-1 composites is not given. What is the ratio between the two?

Reviewer #3 (Remarks to the Author):

In this manuscript, the authors report the synthesis of MOFs with an interesting house-of-cards architecture by orthogonal assembly of single crystal guest MOF nanosheets on a cubic host MOF. The structures of house-of-cards MOFs can be controlled from core-shell to hollow architectures with excellent catalytic properties. The materials have been well characterized and the formation mechanism has also been analyzed in detail. Compared to reported MOF nanosheet assemblies, this work realizes a 3D ordered arrangement of MOF nanosheets, which is interesting. This manuscript is also well constructed and written. I recommend acceptance of this work after a few improvements.

- (1) Low-magnification SEM images of HoC-HS-2 and HoC-HS-3 should be provided.
- (2) A scheme for illustrating the synthesis process should be added in SI.
- (3) Except for $1/3\{220\}$, another group of diffraction points should also be indexed.
- (4) More TEM images of Type I and II nanosheets in other regions are recommended to be provided.
- (5) The meanings of green, blue and red dots in Figure 3g should be clarified in the figure caption.
- (6) The legends of e and f in Figure 5 are inconsistent.
- (7) The performance of 4-NP reduction in the absence of catalyst should be added.
- (8) Several important references of MOF nanosheet assemblies should be cited (e.g. 10.1002/smt.d.201800068, 10.1021/jacs.0c02272).

Reviewer #4 (Remarks to the Author):

In their manuscript the authors describe the fabrication of hetero-aggregates, where cubic MOF (HKUST-I) particles are used as a substrate to grow 2D sheets of a Prussian-Blue analogue (PBA). The synthesis process yields "walls" of PBA nanosheets perpendicular to the substrate and orthogonal to each other. "House of Cards", the term used by the authors, is not appropriate. A house of cards consists of cards not only standing perpendicular on a substrate (table) but also contains cards with an orientation parallel to the substrate. (<https://www.youtube.com/watch?v=SEBBj2BIBm8>).

The notion "Wulff's point", a term used by the authors, is unknown to me.

The characterization of the constructs synthesized using the approach proposed by the authors appears to be adequate, it involves XRD, TEM, and BET measurements.

Alltogether, while I find the results interesting, I do not see that the properties of these constructs reaches the level of scientific excitement needed to warrant a publication in Nature

Communications.

Point-to-point response and revisions

Reviewer 1

General comments: This article by Cheng-Zhong Yu et al reports on the heterostructure core-shell MOFs with house of cards architecture. In this paper, the core-shell MOF was synthesized by epitaxial growth in lattice mismatching on 2D nanosheets on 3D template MOF. In addition, the size and density of 2D nanosheets in the composite MOFs were controlled by the amount of H₂O in the synthesis, and hollow structure MOFs were synthesized by etching core MOFs with relatively low stability. Hollow structure nanosheets arrayed by templating effect exhibits improved catalytic performance compared to randomly arrayed nanosheets. This study has been well carried out, and the explanation for strategy of heterostructure MOF with house of cards architecture. However, it remains the concerns on impact and novelty of this paper in this field. Previously, strategy for the synthesis of 2D hollow nanosheet from Core-shell is reported (Dalton Trans., 2021, 50, 8179-8188, Chem. Eng. J., 2019, 370, 666-676 and Chem. Eng. J., 2022, 449, 137780). In addition, some explanation and claims are not fully supported by proper data. Therefore, it seems that an emphasis different from previous papers is needed, and detailed supplementation seems to be necessary. With the current manuscript, it is difficult to recommend the publication in Nature Communications.

Response: We thank Reviewer 1 for the positive comment stating that “This study has been well carried out, and the explanation for strategy of heterostructure MOF with house of cards architecture”. Your concern about the impact and novelty of this paper has encouraged us to further review the novelty of our work in comparison with reported results. The differences of our work from literature have been emphasized in the revised manuscript to demonstrate the novelty of our work more clearly (see details below). In addition, your other comments are also highly valuable and important for us to improve the quality of our work. In the revised manuscript, we have conducted additional experiments with further revisions to address your queries (see details in the responses to your specific comments).

The novelty of our work compared to literature is summarized as follows:

(1) A unique orthogonal assembly of MOF nanosheet into single-crystalline arrays

Controllable nanosheet growth is important in both fundamental and applied research. On a flat surface, a nanosheet can grow perpendicularly or flat (Fig. R1, A), or randomly (Fig. R1, B) on the surface. Even all the nanosheets are grown perpendicularly, their orientations could be random (Fig. R1, C). Thus, it is difficult to establish a single-crystalline nanosheet array, even one individual nanosheet is single-crystalline. In this regard, our work presents the first report of an orthogonal assembly of single-crystalline MOF nanosheets further into a single-crystalline array. As shown in Fig. R1, D, orthogonally arrayed single-crystalline MOF (CuFe PBA) nanosheets are constructed as the shell on a cubic core MOF (HKUST-1) via epitaxial growth, leading to a single-crystalline CuFe PBA nanosheet array on the surface of HKUST-1 with controllable nanosheet densities and sizes. In addition, the interfacial growth of semi-hexagonal MOF nanosheets with the same crystalline structure but two different shapes is also reported for the first time. More notably, by virtue of the unique epitaxial relationship between CuFe-PBA and HKUST-1, our synthesized CuFe PBA nanosheets expose a naturally nonpreferred (111) facet, different from the (001) dominated ones in previous reports (J. Am. Chem. Soc. 2013, 135, 7, 2793-2799, Nano Lett. 2017, 17, 8, 4958-4963 and Nano Lett. 2018, 18, 6, 4015-4022).

Fig. R1. Schematic diagram of growth orientation of nanosheets on a flat substrate.

In the references mentioned by Reviewer 1, nanosheet arrays composed of MOF derived NiCo alloy@NiCo-sulfide nanoparticles on carbon fibers (Chem. Eng. J., 2019, 370, 666) and Ni-MOF on Co₃O₄ core (Dalton Trans., 2021, 50, 8179) were synthesized by sequential hydrolysis-pyrolysis-sulfidation and solvothermal treatment, respectively. Due to the lack of epitaxial relationship between non-MOF core and MOF shell, the nanosheets were randomly arranged on substrates. In the Ni-HHTP@UiO-66-NH₂ work (Chem. Eng. J., 2022, 449, 137780), a non-epitaxial growth approach was reported for the deposition of randomly distributed Ni-HHTP nanorods on the surface of UiO-66-NH₂. Therefore, the synthetic strategies and resultant structures in three references mentioned by Reviewer 1 are different from our work. The first two references about MOF nanosheet arrays mentioned by Reviewer 1 have also been cited as ref. 19 and 20 in the Introduction part.

We have also carefully re-reviewed all reported works on the growth of MOF nanosheets, whose synthetic strategies and resultant structures can be divided into two categories: (1) non-epitaxial growth of MOF nanosheets without regular spatial arrangement (e.g., the first two references mentioned by Reviewer 1, Nat. Commun. 2017, 8, 15341); (2) epitaxial growth of densely and parallelly stacked MOF nanosheets on the surface of another MOF (e.g., J. Am. Chem. Soc. 2020, 142, 19, 8953), where low-symmetrical MOF substrates were selected for inducing the anisotropic growth of secondary MOF nanosheets along only one direction. In all

relevant reports to interfacial MOF nanosheet growth, there is no evidence that nanosheet array with a single-crystalline nature is observed.

The key innovation in our design is the utilization of a high-symmetrical MOF (i.e., HKUST-1 with a space group of $Fm\bar{3}m$) with a uniform cube morphology as the substrate. The 6 exposed {001} facets are identical. On each facet, the epitaxial growth of CuFe PBA nanosheets occurs at two orthogonal directions, either parallel or perpendicular to the edges of {001} surfaces. This allows orthogonally assembled MOF nanosheets into a single-crystalline framework with large pores in the 3D framework.

Even considering all types of nanosheets, the orthogonal nanosheet array with a single-crystalline nature has only been reported in few zeolite materials (Science, 2012, 336, 1684; Nat. Commun. 2014, 5, 4262; J. Am. Chem. Soc. 2022, 144, 6270). As a new MOF-on-MOF structure, understanding how to assemble low-dimensional MOF nanosheets into 3D and single-crystalline array provides new knowledge and may open a new direction in the MOF field.

(2) Excellent performance with new knowledge

In the revised manuscript, we have conducted oxygen evolution reaction (OER), a core reaction in various energy conversion and storage technologies, as a new application. Based on the new results (see details in response to your comment 10), we believe that the scientific significance and novelty of our work has been greatly improved from the performance perspective. The main observations and conclusions are as follows:

The H-ONSA exhibited a low overpotential of 245 mV at a current density of 10 mA cm⁻², low Tafel slope of 52.1 mV dec⁻¹ and excellent stability (current decay of 7.4% after 24 h). The OER performances are better than conventional (001) exposed CuFe-PBA nanosheet, a commercial noble metal (RuO₂) and most reported MOFs or other transition metal-based OER catalysts. Theoretical calculation indicates that compared to CuFe PBA nanosheets with (100) exposed facets, those with (111) exposed facets offer abundant unsaturated binding sites and stabilize the intermediate species via hydrogen bonding in oxygen evolution reaction (OER). Together with the promoted active site exposure and mass transfer by macroporous array of nanosheets, the OER performance of H-ONSA is enhanced than conventional CuFe PBA

nanosheets. To our knowledge, this is the first report of a new composition (CuFe PBA) and a new structure (orthogonal nanosheet array) in the application of electrocatalysis. The superiority of a naturally nonpreferred (111) facet of CuFe-PBA in electrocatalysis is also unveiled for the first time.

To directly answer to your comment “Therefore, it seems that an emphasis different from previous papers is needed, and detailed supplementation seems to be necessary”, the following clarifications have been added in Page 12 and 16 in the revised manuscript:

"In reported CuFe PBA materials (26, 27), (001) facets are the most exposed surfaces, but nanosheets with (111) dominant facet exposure are rare. To understand the underlying reason, the surface energies of CuFe PBA (111) and (001) surfaces were evaluated by density functional theory (DFT) calculations. The results show that the surface energy of (001) surface ($0.19 \text{ J}\cdot\text{m}^{-2}$) is $0.17 \text{ J}\cdot\text{m}^{-2}$ lower than that of (111) facet ($0.36 \text{ J}\cdot\text{m}^{-2}$). This explains why nanosheets with (001) dominate facet exposure were formed during solution synthesis. In our work, however, the formation of CuFe PBA nanosheets on HKUST-1 surface is restricted by the epitaxial relationship rather than determined by thermodynamics, thus CuFe PBA nanosheets with unusual (111) dominant facet exposure are obtained (Fig. 3)."

"The reports on interfacial growth of MOF nanosheets can be approximately divided into two categories: (1) non-epitaxial growth of MOF nanosheets without regular spatial arrangement (19, 20, 40, 41); (2) epitaxial growth of densely stacked MOF nanosheets along one direction on a low-symmetrical MOF substrate (42). The key difference in our design is the utilization of a high-symmetrical MOF (HKUST-1 cubes with a space group of $Fm\bar{3}m$) as the substrate with six identical exposed $\{001\}$ facets. On each facet, the epitaxial growth of CuFe PBA nanosheets occurs at two orthogonal directions, either parallel or perpendicular to the edges of $\{001\}$ surfaces. This allows orthogonally assembled MOF nanosheets into a single-crystalline array with large pores in the 3D framework, which is rarely reported in MOF-based nanosheet assemblies. Moreover, the MOF-based ONSA is also distinctive from self-pillared zeolite orthogonal nanosheet array formed by self-branching, usually without a uniform morphology (22-25)."

Comment 1: To generalize this strategy, I recommend to replace the word “house of cards architecture”. “House of cards” is an idiom which has another meaning: an unstable structure. Therefore, the expression of “Rigid house of cards architecture” on line 21 of page 4 gives the contradictory impression to readers.

Response: We thank Reviewer 1 for the valuable comment. According to your suggestion, the word "house of cards architecture" has been replaced by "orthogonal nanosheets with single-crystalline arrays (ONSA)".

Comment 2: In the abstract, the expression of “guest MOF” and “host MOF” is not intuitively acceptable based on the common sense regarding to host-guest chemistry. Thus, I recommend to replace the words with “shell MOF” and “core MOF”.

Response: We acknowledge Reviewer 1’s thoughtful comment. Following your suggestion, the words "guest MOF" and "host MOF" have been replaced by "shell MOF" and "core MOF", respectively.

Comment 3: In the Figure 1G, the XRD pattern of nanosheets in HoC-HS-1 is shifted to lower angles compared with the simulated pattern. The explanation is needed.

Response: We appreciate Reviewer 1’s insightful comment. Through careful survey of the crystal structure database, we found a perfectly matched simulated pattern (PDF 86-0513) with our experimental result, which has been updated in the revised Figure 1G in the revised manuscript (see below).

Fig. 1. Characterization of ONSA-HS-1. (A-C) SEM, (D, E) TEM, (F) HAADF-STEM and corresponding element mapping images, and (G) XRD pattern of ONSA-HS-1; (H) FTIR spectra of $K_3[Fe(CN)_6]$, HKUST-1 and ONSA-HS-1.

Comment 4: The discussion on N_2 sorption isotherm is not appropriate. By considering the material ratio of HKUST-1 and CuFe PBA in HoC-HS-1, is the isotherm of HoC-HS-1 plausible? Because CuFe PBA is nonporous, HoC-HS-1 should show the lower uptake than the current result. In addition, authors claim that the large hysteresis loop in high relative pressure ranges from 0.9 to 1.0 is attributed to the macropores by the nanosheets array. First of all, this hysteresis is hard to say large, and there is no supporting evidence for the existence of macropores. In order to confirm this, I recommend to analyze the macropores generated by the arrayed nanosheets on the core MOFs by using TEM and/or SEM.

Response: We acknowledge Reviewer 1's useful comments. By comparison with the literatures, we found that the BET surface area of HKUST-1 ($917 \text{ m}^2 \text{ g}^{-1}$) was lower than the reported results ($1100 \text{ m}^2 \text{ g}^{-1}$, NPG Asia Mater 2020, 12, 58). In this regard, we have retested the N_2 sorption isotherms of both HKUST-1, showing a BET surface area of $1037 \text{ m}^2 \text{ g}^{-1}$ (see below). On this basis, the specific surface area of HoC-HS-1 ($870 \text{ m}^2 \text{ g}^{-1}$) is slightly higher than the value ($771 \text{ m}^2 \text{ g}^{-1}$) calculated by the mass ratio of HKUST-1 and CuFe PBA in HoC-HS-1. This may be contributed by the formation of nanosheets with higher surface area than particles. The similar observation has also been widely observed in previous reports (ACS Sustainable Chem. Eng. 2019, 7, 11564-11570). Additionally, the higher N_2 uptake for ONSA-HS-1 is observed at a relatively high P/P_0 range (> 0.9), indicative of large pores generated by the orthogonal arrangement of CuFe PBA nanosheets. To further support this argument, according to your suggestion, the histogram of pore size distribution obtained from the SEM images is provided (fig. S8, see below), showing an average pore size of $\sim 125 \text{ nm}$. In addition, the description of "large hysteresis loop" has been revised into "hysteresis loop".

Fig. S7. (A) Nitrogen adsorption-desorption isotherms and (B) pore-size distribution curves of HKUST-1, CuFe PBA and ONSA-HS-1.

Fig. S8. Histogram of pore size distribution determined by SEM images in Fig. 1B.

Further clarifications have been added in Page 7 in the revised manuscript as follows:

"The N₂ sorption isotherms of HKUST-1 (fig. S7A) show a type I hysteresis loop with a sharp condensation step at low relative pressure ($P/P_0 < 0.05$), indicating the micropore-dominant nature. By incorporating nonporous CuFe PBA, the N₂ uptake at low relative pressure ($P/P_0 < 0.05$) for ONSA-HS-1 is decreased. The N₂ sorption isotherms of ONSA-HS-1 also exhibit a capillary condensation step at high P/P_0 range (> 0.9) with a hysteresis loop, indicative of large pores generated by the orthogonal arrangement of CuFe PBA nanosheets (34). Then, according to the Brunauer-Emmett-Teller (BET) specific surface area of bare HKUST-1 ($1037 \text{ m}^2 \text{ g}^{-1}$) and CuFe PBA ($53 \text{ m}^2 \text{ g}^{-1}$), the specific surface area ($870 \text{ m}^2 \text{ g}^{-1}$) of HoC-HS-1 is close to the calculated value ($771 \text{ m}^2 \text{ g}^{-1}$) based on mass ratio of HKUST-1 and CuFe PBA in HoC-HS-1. The additional specific surface area in our experimental value may be attributed to the formation of CuFe PBA nanosheets and the orthogonal arrangement. The total pore volume of ONSA-HS-1 ($0.77 \text{ cm}^3 \text{ g}^{-1}$) is higher than that of HKUST-1 ($0.57 \text{ cm}^3 \text{ g}^{-1}$) and CuFe PBA ($0.07 \text{ cm}^3 \text{ g}^{-1}$, fig. S7B and Table S1), also suggesting the formation of additional pores. The average diameter of the large pores is determined to be $\sim 100 \text{ nm}$ from the pore-size distribution curves in fig. S7B, slightly smaller than the value ($\sim 125 \text{ nm}$) measured by the SEM images (fig. S8)."

Comment 5: In the page 7, the explanation of ultrasonication treatment is insufficient. Therefore, I recommend to add more details in the supplementary information.

Response: Following Reviewer 1's kind suggestion, the details of ultrasonication treatment have been added in supplementary as follows.

"Ultrasonic treatment of ONSA-HS-1:

1 mg of ONSA-HS-1 was immersed into 1 mL of ethanol. By ultrasonication treatment at 40 kHz for 2 h, individual CuFe PBA nanosheets were peeled from ONSA-HS-1 particles. The product was then collected by centrifugation and washed with water and ethanol."

Comment 6: In the page 7 to 8, I recommend that the description of calibrating the camera

length with gold nanoparticles as the reference is moved to the supporting information.

Response: We thank Reviewer 1 for the useful comments. The corresponding description has been moved to the experimental section in the supporting information.

Comment 7: In the page 8, There is not enough information to claim that Type 1 is dominant over Type 2. It should be explained with supporting evidences.

Response: We thank Reviewer 1 for the insightful comment helping us to improve the scientific rigor of our work. As can be seen from the original Figs. 2A-D in the manuscript (see below), there is little difference in the intensity of six diffraction spots in an individual Type I or Type II nanosheet. However, for the ED pattern of ONSA-HS-1 from a selected area of a collection of nanosheets (white circle in Fig. 2E), the obtained diffraction spots can be indexed as two superimposed $[1\bar{1}\bar{1}]$ zone-axis single-crystal ED patterns with a relevant 30° rotation (Fig. 2F), consistent with the superposition of diffraction spots of Type I and Type II nanosheets. It is noted that the diffraction spots associated with Type I has stronger intensity than that with Type II. Moreover, this observation is consistent in the SAED patterns from other areas, as shown in fig. S12. As a semi-quantitative measure, the SAED spot intensity was quantified using a software ImageJ (see also below). The SAED spot intensity ratio of Type I versus Type II was calculated to be 9.92 ± 1.43 (fig. S13). Considering both Type I and II nanosheets have similar sizes, their ED spot intensity ratio can be used as a rough estimation of the content difference. From the above analysis, Type I nanosheets are dominant in ONSA-HS-1 over Type II nanosheets.

Fig. 2. Growth orientation of nanosheets in ONSA-HS-1. TEM images (A, C, E) and corresponding SAED patterns (B, D, F) of detached CuFe PBA nanosheet with Type I structure (A, B), Type II structure (C, D) and CuFe PBA nanosheets area in ONSA-HS-1 (E, F), respectively.

Fig. S12. The SAED spot intensity ratio of Type I versus Type II nanosheets in ONSA-HS-1. The three locations were adapted from Fig. 2F, fig. S12B and fig. S12D, respectively.

The following changes have been added in the description of SAED analysis (page 9).

"It is noted that the diffraction spots associated with Type I have stronger intensity than that with Type II. Moreover, this observation is consistent in the SAED patterns from nanosheet locations of other ONSA-HS-1, as shown in fig. S12. By quantifying the spot intensity using a software ImageJ, the SAED spot intensity ratio of Type I versus Type II was calculated to be 9.92 ± 1.43 (fig. S13). Considering both Type I and II nanosheets have comparable sizes, their ED spot intensity ratio can be used as a rough estimation of their content difference. Therefore, Type I nanosheets are dominant in ONSA-HS-1 over Type II, in accordance with both SEM and TEM observations."

Comment 8: Despite the crystallographic studies, the interfacial structure between nanosheets and template MOF is still unclear. Is it expected that the chemical bonding exists? If so, how can be chemical bonding formed even though these MOFs have the lattice mismatching?

Response: We appreciate Reviewer 1's insightful comments. We agree with your comment that the interfacial structure between CuFe PBA and HKUST-1 is an important question. In our work we tried to answer to this question indirectly. The XPS spectrum of HKUST-1 (A-C, see below) indicates the existence of unsaturated Cu atoms on the surface. The unsaturated Cu atom ratio increased with the amount of added water in this activation step, in accordance with a literature report (Chemphyschem 2012, 13, 2025-2029). With increased unsaturated Cu atom ratio, higher density of nucleation sites and finally nanosheets was observed (D-F, see also below). This observation indicates that the unsaturated Cu atoms may serve as the bonding sites to connect with $[\text{Fe}(\text{CN})_6]^{3-}$ for inducing the nucleation and growth of CuFe PBA. The similar role of Cu atoms for the interfacial growth of MOF-5 via forming $\text{Cu}_{(\text{HKUST-1})}\text{-O}_{(\text{MOF-5})}$ bonds has also been report (Nat Commun 2019, 10, 3620).

We understand the above results are not direct evidence. Nevertheless, the interfacial bonding structure, presumably Cu-C-N-Fe as we deduced in the above paragraph, exists also in CuFe PBA, making it difficult to differentiate the interfacial structure (e.g., using XPS spectra, TEM, etc).

High-resolution XPS spectra of Cu 2p of HKUST-1 after activation by (A) 0, (B) 1 and (C) 3 mL of water (A, B and C correspond to figs. S18A, B and C, respectively). SEM images of HoC-HS-3 (D), HoC-HS-2 (E) and HoC-HS-1 (F) collected after growth of CuFe PBA for 10 min (D, E and F correspond to figs. S20E, S20A and S9A, respectively).

We agree with you that HKUST-1 and CuFe PBA have lattice mismatching (6.16%, 8.42% and 10.66%). According to a previous report (Nano Lett. 2014, 14, 3, 1526-1529), for two MOFs with lattice mismatch of even 26%, the chemical bonds can be formed at the epitaxial interface because the relatively flexible framework of MOFs allows the structural deformation with generation of misfit dislocation to adapt for the lattice mismatch (Nature Mater 2004, 3, 87-90; Chem. Sci., 2020, 11, 3680-3686).

The following revisions have been added in Page 15 in the revised manuscript.

"The coordination unsaturated Cu sites with higher activity are anticipated to act as the nucleation sites to bond with $[\text{Fe}(\text{CN})_6]^{3-}$ for inducing the growth of CuFe PBA. The similar role of Cu atoms on the $\{100\}$ facets of HKUST-1 has also been reported for the interfacial growth of MOF-5 via forming $\text{Cu}_{(\text{HKUST-1})}\text{-O}_{(\text{MOF-5})}$ bonds (39)."

Comment 9: In Figure 4, it is necessary to state the kinds of metal in the EDS mapping data (figure 4e-g, 4m-o). Also, it would be better to match the schematic image of figure 4a, i with the color of the metal used for mapping (e.g. in figure 4a, host is purple and guest is green)

Response: We are grateful for Reviewer 1's valuable comments. The metals in the EDS mapping images (Figs. 4E-G, 4M-O) have been marked and their colors have also been changed to match the schematic image as below.

Fig. 1. Characterization of ONSA-HC-1. (A-C) SEM, (D, E) TEM, (F) HAADF-STEM and corresponding element mapping images, and (G) XRD pattern of ONSA-HS-1; (H) FTIR spectra of $\text{K}_3[\text{Fe}(\text{CN})_6]$, HKUST-1 and ONSA-HS-1.

Fig. 4. Controllable synthesis of ONSA by regulating activation degree. (A, I) Structural illustration, (B, J) SEM, (C, K) TEM, (D, L) STEM and corresponding elements mapping (E-G, M-O) images and line scan patterns (H, P) of ONSA-HS-2 and 3.

Comment 10: To claim the good catalytic performance of the hollow structure, it is need to provide the result of catalytic reaction test for core-shell.

Response: We appreciate Reviewer 1's comments. In the revised manuscript, we have conducted alkaline oxygen evolution reaction (OER, a core reaction in various energy conversion and storage technologies) as a new application according to the comments from Editor, Reviewers 2 and 4. However, considering that the HKUST-1 is instable in alkaline solution, the performance of CuFe PBA@HKUST-1 core-shell hybrids was not measured. Instead, the superiority of CuFe PBA H-ONSA was demonstrated using four control groups: nanosheets-001, nanosheets-111 and nanoparticles of CuFe PBA, and a commercial noble

metal-based catalyst (RuO_2). Benefiting from the unique structural advantages, the H-ONSA exhibited enhanced OER activity than control samples and even RuO_2 . The results have been added as Figs. 6 and 7, figs. S22-26 in the revised manuscript (see also below).

Fig. 6. OER performance of H-ONSA. (a) Structural advantages of H-ONSA for OER; (b) LSV curves, (c) overpotentials required for current density of 10 mA cm^{-2} , (d) Tafel slopes from LSV curves; (e) CV current density versus scan rate of different catalysts; the linear slope is equivalent to the double-layer capacitance (Cdl); (f) EIS spectra recorded at 1.479 V versus RHE; (g) LSV curves before and after successive CV scanning; (h) long-term stability of H-ONSA and RuO_2 .

Figure 7. Optimized structures of the reaction intermediates on the CuFe PBA (111) surface (A) and (001) surface (B). (C) Free energy reaction profiles of the OER process at 1.23 V vs SHE.

Fig. S22. (A) TEM images and (B) XRD pattern of collapsed CuFe PBA nanosheets.

Fig. S23. (A, B) SEM images, (C) TEM image and corresponding (D) SAED pattern of CuFe PBA nanosheets-001.

Fig. S24. XRD pattern of CuFe PBA nanosheets-001.

Fig. S25. (A) SEM and (B) TEM images, and (C) XRD pattern of H-ONSA after cycling.

Fig. S26. CV curves between 1.35 and 1.45 V vs. RHE for (A) Nanocrystals, (B) Nanosheets, (C) H-ONSA and (D) CuFe PBA nanosheets-001.

Corresponding descriptions have been added in Page 17 in the revised manuscript as follows:

"Electrocatalytic OER performance

Oxygen evolution reaction (OER) plays important roles in various oxygen-involved energy conversion and storage technologies, where the development of effective electrocatalysts is the key (43, 44). Herein, the OER performances of H-ONSA were evaluated in 1.0 M KOH electrolyte with CuFe PBA nanocrystals, peeled CuFe PBA nanosheets (denoted as nanosheets-111, fig. S22) and commercial RuO₂ catalyst. In addition, to explore the effect of facet exposure, (001) dominated CuFe PBA nanosheets (denoted as nanosheets-001, fig. S23 and 24) were also synthesized. Fig. 6B shows the linear sweep voltammetry (LSV) curves of all samples. The H-ONSA requires lowest overpotential of 258 mV to reach the current density of 10 mA cm⁻², superior to those of CuFe PBA nanocrystals (430 mV), nanosheets-001 (350 mV), nanosheets-111 (301 mV) and even commercial RuO₂ (258 mV), indicating the highest OER activity (Fig. 6C). In addition, the OER kinetics of four catalysts were examined by Tafel plots (Fig. 6D). The H-ONSA exhibits a lower Tafel slope of 52.1 mV dec⁻¹ than nanocrystals (115.7 mV dec⁻¹), nanosheets-001 (96.0 mV dec⁻¹), nanosheets-111 (94.3 mV dec⁻¹) and RuO₂ (67.5 mV dec⁻¹), showing accelerated OER kinetics. Notably, the remarkable OER performances of H-ONSA is comparable to or even higher than most reported MOFs or other transition metal-based OER catalysts (Table S2).

Apart from the activity, stability is also crucial for OER electrocatalysts. Successive CV with a

scan rate of 50 mV s^{-1} was carried out. After 500 cycles, the obtained LSV curve of H-ONSA is similar to the initial one with slight increase of the overpotential (Fig. 6G), displaying a reasonable stability. The long-term durability of H-ONSA was further assessed by the chronoamperometric measurements (Fig. 6H). After 24 h, only about 7.4 % of current decay is observed in H-ONSA, significantly smaller than that of RuO_2 (high up to 44.5 %), suggesting the excellent OER stability. The sample after cycling test was also characterized by TEM, SEM and XRD, showing well-preserved morphology and crystalline structure (fig. S25).

To understand the superior OER performance of H-ONSA, the electrochemical surface area (ECSA) is further studied by calculating double-layer capacitances (C_{dl}) (45). The CV curves of different samples at different scan rates are presented in fig. S26. As expected, the H-ONSA affords more available active sites with a C_{dl} value of 34.8 mF cm^{-2} (Fig. 6E), which is ~ 9.9 , 3.5 and 3.1 times higher than that of nanocrystal (3.5 mF cm^{-2}), nanosheets-001 surface (9.97 mF cm^{-2}) and nanosheets-111 (11.3 mF cm^{-2}). Moreover, electrochemical impedance spectroscopy (EIS) was used to investigate the charge transport kinetics. The EIS spectra (Fig. 6F) show the lowest charge transfer resistance (R_{ct}) of H-ONSA among all samples, revealing the facilitated charge transfer.

To gain further insight into the facet exposure and OER activity relationship, DFT calculations were carried out to simulate the OER processes over (111) and (001) surfaces of CuFe PBA. The optimized structures of the OER intermediates on (111) and (001) surfaces are shown in Fig. 7, A and B, respectively. The corresponding free energy reaction profiles at $U = 1.23 \text{ V vs. SHE}$ are shown in Fig. 7C. In line with the experimental observation that CuFe PBA nanosheets with a (111) dominant surface exhibit enhanced OER performance, the accumulated free energy cost of OOH_{ad} generation, a rate-determining step during OER process (46,47), on the (111) surface is 1.10 eV, which is 0.26 eV lower than that on the (001) surface. The calculation results indicate that the superior OER activity of (111) surface can be attributed to two aspects. Firstly, the Fe atoms on the (111) surface are unsaturated and five-coordinated (as indicated by the red arrows in Fig. 7A), whereas on the (001) surface, they are saturated and six-coordinated (red arrows in Fig. 7B). As a result, the water molecules can directly adsorb onto the Fe sites on the (111) surface, while the replacement of -CN group is necessary for the water adsorption on the (001) surface. Secondly, on the (111) surface, the nearby -CN groups can stabilize the hydrogen-

containing intermediate species adsorbed on the surface via the formation of hydrogen bonds. Particularly, the formation of a strong hydrogen bond (with a bond length of 1.39 Å) leads to a reduced energy cost of the rate-determining $*O \rightarrow *OOH$ step on the (111) surface (0.40 eV), which is 0.15 eV lower than the energy cost on (100) surface (0.55 eV) in the absence of hydrogen bond formation.

Based on the above results, the enhanced OER performance of H-ONSA should be ascribed to the following structural superiorities (Fig. 6A). (1) The (111) dominated nanosheets offer unsaturated active sites and stabilize hydrogen-containing intermediate species, favoring the OER process. (2) The orthogonal arrangement of free-standing nanosheets reduces the likelihood of restacking of nanosheets, providing abundant surface active sites and thus improving the catalytic activity. (3) The created vertical and penetrating pore channels (48, 49), acting like a "highway", facilitates the diffusion of electrolytes and O_2 molecules. (4) The hollow structure is also conducive for the active site utilization and mass transfer (50, 51), enhancing the electrocatalytic properties."

Comment 11: Please revise and update any errors and typos in the papers.

- (1) In the Figure 3, (a) (110) plane and (b) {001} plane of HKUST-1 should be replaced to (110) plane of CuFe PBA and (001) plane of HKUST-1.
- (2) In the Figure 4, (d, l) STEM and corresponding elements mapping (e-j, m-o) should be replaced to (d, l) STEM and corresponding elements mapping (e-g, m-o).
- (3) In the page 14, the period placed between morphology and (18-21) should be removed.
- (4) In the figure S5B, the unit of X axis is an error.
- (5) In the reference 19, the page number of the reference paper is an error. (1-9 is changed to 4262)
- (6) In the reference 29, the page number of the reference paper is an error. (1122279 is changed to 112279)

Response: We thank Reviewer 1 for pointing out our mistakes. We have carefully revised the manuscript as follows.

- (1) The sentence "(A) (110) plane and (B) {001} plane of HKUST-1" has been replaced by "(A) (110) plane of CuFe PBA and (B) (001) plane of HKUST-1" in Fig. 3.

- (2) The sentence “(D, L) STEM and corresponding elements mapping (E-J, M-O)” has been replaced to “(D, L) STEM and corresponding elements mapping (E-G, M-O)” in Fig. 4.
- (3) The period between morphology and (18-21) ((22-25) in the new version) has been removed.
- (4) The unit of X axis has been corrected to “Pore Width (nm)” In the fig. S5B (fig. 7B in the new version).
- (5) The page number of reference paper 19 (reference 23 in the new version) has been corrected.
- (6) The page number of reference paper 29 (reference 35 in the new version) has been revised.

Reviewer 2

General comments: The manuscript describes the growth of CuFe Prussian Blue Analogue (PBA) nanosheets on micron scale cubic crystals of the Cu MOF HKUST-1, with the source of Cu being the MOF itself. The morphology of the composite can be controlled, changing the height, density and arrangement of the PBA nanosheets, and the interior HKUST-1 MOF can be selectively etched to leave a free-standing hollow structure. The catalytic properties of the different morphologies of the CuFe PBA material – nanosheets, nanocrystals, and hollow “house of cards” cubes – are assessed and results interpreted in terms of the accessibility of the substrate for catalytic sites in the different morphologies.

The manuscript is essentially based around discussion of an interesting and visually pleasing material, with large amounts of content devoted to describing the morphology and rationalizing its formation. I found the “house-of-cards” description to be meaningless; it seems like an attempt to dress up a material for publication by comparison with a macroscale object. The assembly looks nothing like a house of cards (which has at least short-range order), it is simply a random arrangement of sheets growing off a cubic core with some orientational preference. I would suggest removing this unnecessary comparison.

I also found the description of type I and type II nanosheets difficult to interpret when it first arises in the manuscript. The image in Figure 1e is not clear at all – it is impossible to tell the difference – whereas the schematic in Figure 2 is much clearer. What is not clear is why there is a need to make this distinction. The only place where convincing evidence of two different “forms” is provided is in the characterization of individual sheets cleaved from the MOF

support. Do we know if these “types” are nanosheets growing in slightly different directions from the MOFs, or simply parts of sheets broken away from the MOF in different places? Some much clearer microscopy of the conjugates would be required to directly evidence this. In any case, I see no reason to make this distinction, which is simply based upon how the nanosheets project from the MOF, other than to facilitate a more lengthy discussion of the material structure. The various PBA morphologies are used as catalysts for the reduction of p-nitrophenol with sodium borohydride. This is a very standard reaction which can be catalyzed by a huge range of materials, to the extent that someone has even written a review on the topic (DOI: 10.1007/s40089-021-00362-w, not cited). Why do the PBA particles stand out here? What is the comparison with alternative systems in terms of catalyst loading, kinetics, etc? It is well known that greater surface area enhances catalytic activity, so the results with the different morphologies of the PBA materials are not at all surprising.

In addition, the reporting of the catalysis experiments could be much improved. The mol % of the catalyst is not reported, and it is described as being added as “an aqueous solution”. If these materials have dissolved fully, then surely they are no longer in the desired morphology? The authors should report both the characterization of the material after catalysis, and the Cu and Fe content of the supernatants to estimate degradation. It could simply be that the hollow cube morphology, which has the highest surface area, degrades more than the other morphologies and releases catalytic metal ions or metal oxide nanoparticles into solution (iron oxide nanoparticles are known to catalyze this reaction).

Overall, while the materials are interesting to look at and characterised in reasonable detail, the manuscript does not provide any noteworthy new results. This is simply an interesting material, which is discussed in depth and used for very rudimentary catalysis work (for which significant additional work is required), and so I cannot recommend publication in Nature Communications.

Response: We thank Reviewer 2 for the positive comments on our designed materials, stating that "The manuscript is essentially based around discussion of an interesting and visually pleasing material". Your queries of the novelty and frank criticism on catalysis part have encouraged us to review the true novelty of our work and to explore the performance of our material in oxygen evolution reaction. We have made substantial revisions in the revised

manuscript to fully address your comments by adding new experimental results and clarifications (see details below). We feel your critical comments are highly valuable and important for us to improve the quality of our work, which makes the novelty of this story clearer to the readers. The specific revisions and responses to your comments are as follows:

(1) We thank Reviewer 2 for the useful comments. Following your thoughtful suggestions, the word "house of cards" has been removed and replaced by "orthogonal nanosheets with single-crystalline arrays (ONSA)".

(2) For your comments in the third paragraph, it contains several questions. We will answer to them separately.

(a) The first concern is mainly about the characterizations of two types of nanosheets and the necessity to distinct them.

Response: First, SEM images of ONSA-HS-1 with higher resolution have been provided (see below), from which two types of nanosheets with different shapes can be clearly distinguished with Type I dominant than Type II. Both the two types of nanosheets are well arranged, either parallel or perpendicular to the edges of HKUST-1 cube. In addition, the diffraction patterns of the nanosheets in ONSA-HS-1 (Fig. 2, see below) can be indexed as two superimposed $[1\bar{1}\bar{1}]$ zone-axis single-crystal ED patterns with a relevant 30° rotation, which is consistent with the superposition of diffraction spots of Type I and Type II nanosheets. As a semi-quantitative measure, the SAED spot intensity was further quantified using a software ImageJ (see also below). The SAED spot intensity ratio of Type I versus Type II was calculated to be 9.92 ± 1.43 (fig. S13). Considering both Type I and II nanosheets have similar sizes, their ED spot intensity ratio can be used as a rough estimation of the content difference. Based on this analysis, Type I nanosheets are dominant in ONSA-HS-1 over Type II nanosheets. Collectively, the above results verify that the ONSA-HS-1 is composed of two types of nanosheets with different orientations rather than sheet fractures in different places. Notably, such a single-crystalline nanosheet array orthogonally assembled by two types of single crystal nanosheets in our work

has never been observed in previous nanosheet assemblies. Therefore, we believe that the distinction and discussion of the two types of PBA nanosheets is necessary and helpful for in depth understanding the structure and growth mechanism of this new material.

Fig. S4.

The SEM images (A, D) and corresponding enlarged SEM images (B, E) of ONSA-HS-1.

The two typical nanosheets in fig. S4B and E were marked with different colors in C and F, respectively.

Fig. 2. Growth orientation of nanosheets in ONSA-HS-1. TEM images (A, C, E) and corresponding SAED patterns (B, D, F) of detached CuFe PBA nanosheet with Type I structure (A, B), Type II structure (C, D) and CuFe PBA nanosheets area in ONSA-HS-1 (E, F),

respectively.

Fig. S13. The SAED spot intensity ratio of Type I versus Type II nanosheets in ONSA-HS-1. The three locations were adapted from Fig. 2F, fig. S12B and fig. S12D, respectively.

Further clarifications have been added in Page 5 the revised manuscript as follows:

"At higher magnification (Figs. 1, B and C, and figs. S4), two types of semi-hexagonal nanosheets can be seen on the cube surface with an average thickness of ~50 nm. Type I (marked with red line in figs. S4, B and D) is dominant and exposes three sides, and Type II (marked with blue line) is minor with four sides exposed."

"It is noted that the diffraction spots associated with Type I have stronger intensity than that with Type II. Moreover, this observation is consistent in the SAED patterns from nanosheet locations of other ONSA-HS-1, as shown in fig. S12. By quantifying the spot intensity using a software ImageJ, the SAED spot intensity ratio of Type I versus Type II was calculated to be 9.92 ± 1.43 (fig. S13). Considering both Type I and II nanosheets have comparable sizes, their ED spot intensity ratio can be used as a rough estimation of their content difference. Therefore, Type I nanosheets are dominant in ONSA-HS-1 over Type II, in accordance with both SEM and TEM observations. Moreover, the interfacial growth of semi-hexagonal MOF nanosheets with the same crystalline structure but two different shapes is also reported for the first time."

(b) The second concern is about the performance study.

Response: In the revised manuscript, we have conducted alkaline oxygen evolution reaction (OER, a core reaction in various energy conversion and storage technologies) as a new application according to the comments from Editor, Reviewers 2 and 4. However, considering that the HKUST-1 is unstable in alkaline solution, the performance of CuFe PBA@HKUST-1 core-shell hybrids was not measured. Instead, the superiority of CuFe PBA H-ONSA was demonstrated using four control groups: nanosheets-001, nanosheets-111 and nanoparticles of CuFe PBA, and a commercial noble metal-based catalyst (RuO_2). Benefiting from the unique structural advantages, the H-ONSA exhibited enhanced OER activity than control samples and even RuO_2 . The results have been added as Figs. 6 and 7, figs. S22-26 in the revised manuscript (see also below).

Fig. 6. OER performance of H-ONSA. (a) Structural advantages of H-ONSA for OER; (b) LSV curves, (c) overpotentials required for current density of 10 mA cm^{-2} , (d) Tafel slopes from LSV curves; (e) CV current density versus scan rate of different catalysts; the linear slope is equivalent to the double-layer capacitance (Cdl); (f) EIS spectra recorded at 1.479 V versus RHE; (g) LSV curves before and after successive CV scanning; (h) long-term stability of H-ONSA and RuO_2 .

Figure 7. Optimized structures of the reaction intermediates on the CuFe PBA (111) surface (A) and (001) surface (B). (C) Free energy reaction profiles of the OER process at 1.23 V vs SHE.

Fig. S22. (A) TEM images and (B) XRD pattern of collapsed CuFe PBA nanosheets.

Fig. S23. (A, B) SEM images, (C) TEM image and corresponding (D) SAED pattern of CuFe PBA nanosheets-001.

Fig. S24. XRD pattern of CuFe PBA nanosheets-001.

Fig. S25. (A) SEM and (B) TEM images, and (C) XRD pattern of H-ONSA after cycling.

Fig. S26. CV curves between 1.35 and 1.45 V vs. RHE for (A) Nanocrystals, (B) Nanosheets, (C) H-ONSA and (D) CuFe PBA nanosheets-001.

Corresponding descriptions have been added in Page 17 in the revised manuscript as follows:

"Electrocatalytic OER performance

Oxygen evolution reaction (OER) plays important roles in various oxygen-involved energy conversion and storage technologies, where the development of effective electrocatalysts is the key (43, 44). Herein, the OER performances of H-ONSA were evaluated in 1.0 M KOH electrolyte with CuFe PBA nanocrystals, peeled CuFe PBA nanosheets (denoted as nanosheets-111, fig. S22) and commercial RuO₂ catalyst. In addition, to explore the effect of facet exposure, (001) dominated CuFe PBA nanosheets (denoted as nanosheets-001, fig. S23 and 24) were also synthesized. Fig. 6B shows the linear sweep voltammetry (LSV) curves of all samples. The H-ONSA requires lowest overpotential of 258 mV to reach the current density of 10 mA cm⁻², superior to those of CuFe PBA nanocrystals (430 mV), nanosheets-001 (350 mV), nanosheets-111 (301 mV) and even commercial RuO₂ (258 mV), indicating the highest OER activity (Fig. 6C). In addition, the OER kinetics of four catalysts were examined by Tafel plots (Fig. 6D). The H-ONSA exhibits a lower Tafel slope of 52.1 mV dec⁻¹ than nanocrystals (115.7 mV dec⁻¹), nanosheets-001 (96.0 mV dec⁻¹), nanosheets-111 (94.3 mV dec⁻¹) and RuO₂ (67.5 mV dec⁻¹), showing accelerated OER kinetics. Notably, the remarkable OER performances of H-ONSA is comparable to or even higher than most reported MOFs or other transition metal-based OER catalysts (Table S2).

Apart from the activity, stability is also crucial for OER electrocatalysts. Successive CV with a

scan rate of 50 mV s^{-1} was carried out. After 500 cycles, the obtained LSV curve of H-ONSA is similar to the initial one with slight increase of the overpotential (Fig. 6G), displaying a reasonable stability. The long-term durability of H-ONSA was further assessed by the chronoamperometric measurements (Fig. 6H). After 24 h, only about 7.4 % of current decay is observed in H-ONSA, significantly smaller than that of RuO_2 (high up to 44.5 %), suggesting the excellent OER stability. The sample after cycling test was also characterized by TEM, SEM and XRD, showing well-preserved morphology and crystalline structure (fig. S25).

To understand the superior OER performance of H-ONSA, the electrochemical surface area (ECSA) is further studied by calculating double-layer capacitances (C_{dl}) (45). The CV curves of different samples at different scan rates are presented in fig. S26. As expected, the H-ONSA affords more available active sites with a C_{dl} value of 34.8 mF cm^{-2} (Fig. 6E), which is ~ 9.9 , 3.5 and 3.1 times higher than that of nanocrystal (3.5 mF cm^{-2}), nanosheets-001 surface (9.97 mF cm^{-2}) and nanosheets-111 (11.3 mF cm^{-2}). Moreover, electrochemical impedance spectroscopy (EIS) was used to investigate the charge transport kinetics. The EIS spectra (Fig. 6F) show the lowest charge transfer resistance (R_{ct}) of H-ONSA among all samples, revealing the facilitated charge transfer.

To gain further insight into the facet exposure and OER activity relationship, DFT calculations were carried out to simulate the OER processes over (111) and (001) surfaces of CuFe PBA. The optimized structures of the OER intermediates on (111) and (001) surfaces are shown in Fig. 7, A and B, respectively. The corresponding free energy reaction profiles at $U = 1.23 \text{ V vs. SHE}$ are shown in Fig. 7C. In line with the experimental observation that CuFe PBA nanosheets with a (111) dominant surface exhibit enhanced OER performance, the accumulated free energy cost of OOH_{ad} generation, a rate-determining step during OER process (46,47), on the (111) surface is 1.10 eV, which is 0.26 eV lower than that on the (001) surface. The calculation results indicate that the superior OER activity of (111) surface can be attributed to two aspects. Firstly, the Fe atoms on the (111) surface are unsaturated and five-coordinated (as indicated by the red arrows in Fig. 7A), whereas on the (001) surface, they are saturated and six-coordinated (red arrows in Fig. 7B). As a result, the water molecules can directly adsorb onto the Fe sites on the (111) surface, while the replacement of -CN group is necessary for the water adsorption on the (001) surface. Secondly, on the (111) surface, the nearby -CN groups can stabilize the hydrogen-

containing intermediate species adsorbed on the surface via the formation of hydrogen bonds. Particularly, the formation of a strong hydrogen bond (with a bond length of 1.39 Å) leads to a reduced energy cost of the rate-determining $*O \rightarrow *OOH$ step on the (111) surface (0.40 eV), which is 0.15 eV lower than the energy cost on (100) surface (0.55 eV) in the absence of hydrogen bond formation.

Based on the above results, the enhanced OER performance of H-ONSA should be ascribed to the following structural superiorities (Fig. 6A). (1) The (111) dominated nanosheets offer unsaturated active sites and stabilize hydrogen-containing intermediate species, favoring the OER process. (2) The orthogonal arrangement of free-standing nanosheets reduces the likelihood of restacking of nanosheets, providing abundant surface active sites and thus improving the catalytic activity. (3) The created vertical and penetrating pore channels (48, 49), acting like a "highway", facilitates the diffusion of electrolytes and O_2 molecules. (4) The hollow structure is also conducive for the active site utilization and mass transfer (50, 51), enhancing the electrocatalytic properties."

(3) In addition, the reporting of the catalysis experiments could be much improved. The mol % of the catalyst is not reported, and it is described as being added as "an aqueous solution". If these materials have dissolved fully, then surely they are no longer in the desired morphology? The authors should report both the characterization of the material after catalysis, and the Cu and Fe content of the supernatants to estimate degradation. It could simply be that the hollow cube morphology, which has the highest surface area, degrades more than the other morphologies and releases catalytic metal ions or metal oxide nanoparticles into solution (iron oxide nanoparticles are known to catalyze this reaction).

Response: We acknowledge Reviewer 2's helpful comment. To address your concern about the catalysis application, we have conducted oxygen evolution reaction (OER, a core reaction in various energy conversion and storage technologies) as a new application (see our response to your Comment 2d).

(4) Overall, while the materials are interesting to look at and characterised in reasonable detail,

the manuscript does not provide any noteworthy new results. This is simply an interesting material, which is discussed in depth and used for very rudimentary catalysis work (for which significant additional work is required)

Response: We appreciate Reviewer 2's helpful suggestion. To address your concern about the catalysis application, we have conducted oxygen evolution reaction (OER, a core reaction in various energy conversion and storage technologies) as a new application (see our response to your Comment 2d). On this basis, the new findings of our work compared to reported ones are re-summarized from both synthesis, structure and performance perspectives as follows:

The novelty of our work compared to literature is summarized as follows:

(1) A unique orthogonal assembly of MOF nanosheet into single-crystalline arrays

Controllable nanosheet growth is important in both fundamental and applied research. On a flat surface, a nanosheet can grow perpendicularly or flat (Fig. R1, A), or randomly (Fig. R1, B) on the surface. Even all the nanosheets are grown perpendicularly, their orientations could be random (Fig. R1, C). Thus, it is difficult to establish a single-crystalline nanosheet array, even one individual nanosheet is single-crystalline. In this regard, our work presents the first report of an orthogonal assembly of single-crystalline MOF nanosheets further into a single-crystalline array. As shown in Fig. R1, D, orthogonally arrayed single-crystalline MOF (CuFe PBA) nanosheets are constructed as the shell on a cubic core MOF (HKUST-1) via epitaxial growth, leading to a single-crystalline CuFe PBA nanosheet array on the surface of HKUST-1 with controllable nanosheet densities and sizes. In addition, the interfacial growth of semi-hexagonal MOF nanosheets with the same crystalline structure but two different shapes is also reported for the first time. More notably, by virtue of the unique epitaxial relationship between

CuFe-PBA and HKUST-1, our synthesized CuFe PBA nanosheets expose a naturally nonpreferred (111) facet, different from the (001) dominated ones in previous reports (J. Am. Chem. Soc. 2013, 135, 7, 2793-2799, Nano Lett. 2017, 17, 8, 4958-4963 and Nano Lett. 2018, 18, 6, 4015-4022).

Fig. R1. Schematic diagram of growth orientation of nanosheets on a flat substrate.

We have also carefully re-reviewed all reported works on the growth of MOF nanosheets, whose synthetic strategies and resultant structures can be divided into two categories: (1) non-epitaxial growth of MOF nanosheets without regular spatial arrangement (e.g., the first two references mentioned by Reviewer 1, Nat. Commun. 2017, 8, 15341); (2) epitaxial growth of densely and parallelly stacked MOF nanosheets on the surface of another MOF (e.g., J. Am. Chem. Soc. 2020, 142, 19, 8953), where low-symmetrical MOF substrates were selected for inducing the anisotropic growth of secondary MOF nanosheets along only one direction. In all relevant reports to interfacial MOF nanosheet growth, there is no evidence that nanosheet array with a single-crystalline nature is observed.

The key innovation in our design is the utilization of a high-symmetrical MOF (i.e., HKUST-1 with a space group of $Fm\bar{3}m$) with a uniform cube morphology as the substrate. The 6 exposed {001} facets are identical. On each facet, the epitaxial growth of CuFe PBA nanosheets occurs

at two orthogonal directions, either parallel or perpendicular to the edges of {001} surfaces. This allows orthogonally assembled MOF nanosheets into a single-crystalline framework with large pores in the 3D framework.

Even considering all types of nanosheets, the orthogonal nanosheet array with a single-crystalline nature has only been reported in few zeolite materials (Science, 2012, 336, 1684; Nat. Commun. 2014, 5, 4262; J. Am. Chem. Soc. 2022, 144, 6270). As a new MOF-on-MOF structure, understanding how to assemble low-dimensional MOF nanosheets into 3D and single-crystalline array provides new knowledge and may open a new direction in the MOF field.

(2) Excellent performance with new knowledge

In the revised manuscript, we have conducted oxygen evolution reaction (OER), a core reaction in various energy conversion and storage technologies, as a new application. Based on the new results (see details in response to your comment 10), we believe that the scientific significance and novelty of our work has been greatly improved from the performance perspective. The main observations and conclusions are as follows:

The H-ONSA exhibited a low overpotential of 245 mV at a current density of 10 mA cm⁻², low Tafel slope of 52.1 mV dec⁻¹ and excellent stability (current decay of 7.4% after 24 h). The OER performances are better than conventional (001) exposed CuFe-PBA nanosheet, a commercial noble metal (RuO₂) and most reported MOFs or other transition metal-based OER catalysts. Theoretical calculation indicates that compared to CuFe PBA nanosheets with (100) exposed facets, those with (111) exposed facets offer abundant unsaturated binding sites and stabilize the intermediate species via hydrogen bonding in oxygen evolution reaction (OER). Together with the promoted active site exposure and mass transfer by macroporous array of nanosheets, the OER performance of H-ONSA is enhanced than conventional CuFe PBA nanosheets. Together with the facilitated mass diffusion and active site exposure by the orthogonal arrangement of nanosheets that create large pore channels, the OER performance of H-NOSA is significantly reinforced. To our knowledge, this is the first report of a new composition (CuFe PBA) and a new structure (orthogonal nanosheet array) in the application of electrocatalysis. The superiority of a naturally nonpreferred (111) facet of CuFe-PBA in

electrocatalysis is also unveiled for the first time.

To directly answer to your comment “Therefore, it seems that an emphasis different from previous papers is needed, and detailed supplementation seems to be necessary”, the following clarifications have been added in Page 12 and 16 in the revised manuscript:

"In reported CuFe PBA materials (26, 27), (001) facets are the most exposed surfaces, but nanosheets with (111) dominant facet exposure are rare. To understand the underlying reason, the surface energies of CuFe PBA (111) and (001) surfaces were evaluated by density functional theory (DFT) calculations. The results show that the surface energy of (001) surface ($0.19 \text{ J}\cdot\text{m}^{-2}$) is $0.17 \text{ J}\cdot\text{m}^{-2}$ lower than that of (111) facet ($0.36 \text{ J}\cdot\text{m}^{-2}$). This explains why nanosheets with (001) dominate facet exposure were formed during solution synthesis. In our work, however, the formation of CuFe PBA nanosheets on HKUST-1 surface is restricted by the epitaxial relationship rather than determined by thermodynamics, thus CuFe PBA nanosheets with unusual (111) dominant facet exposure are obtained (Fig. 3)."

"The reports on interfacial growth of MOF nanosheets can be approximately divided into two categories: (1) non-epitaxial growth of MOF nanosheets without regular spatial arrangement (19, 20, 40, 41); (2) epitaxial growth of densely stacked MOF nanosheets along one direction on a low-symmetrical MOF substrate (42). The key difference in our design is the utilization of a high-symmetrical MOF (HKUST-1 cubes with a space group of $Fm\bar{3}m$) as the substrate with six identical exposed $\{001\}$ facets. On each facet, the epitaxial growth of CuFe PBA nanosheets occurs at two orthogonal directions, either parallel or perpendicular to the edges of $\{001\}$ surfaces. This allows orthogonally assembled MOF nanosheets into a single-crystalline array with large pores in the 3D framework, which is rarely reported in MOF-based nanosheet assemblies. Moreover, the MOF-based ONSA is also distinctive from self-pillared zeolite orthogonal nanosheet array formed by self-branching, usually without a uniform morphology (22-25)."

(5) Some other minor suggestions for improving the manuscript from Reviewer 2:

Comment 1. It would be useful to provide structural composition of the two materials in the introduction for the general reader.

Response: We thank Reviewer 2 for the constructive suggestion. The structural composition of HKUST-1 and CuFe PBA have been provided in the introduction as follows:

"The synthesis of orthogonal nanosheets with single-crystalline arrays (ONSA) involves the oriented growth of two types of CuFe PBA (PBA=prussian blue analogues, space group of $Fm\bar{3}m$, $a=10.10$ Å, molecular formula: $Cu_3[Fe(CN)_6]_2$) nanosheets as the shell on the surface of cubic HKUST-1 (HKUST=Hong Kong University of Science and Technology, space group of $Fm\bar{3}m$, $a=26.34$ Å, molecular formula: $C_{18}H_{12}Cu_3O_{15}$) as the core. "

Comment 2. The actual composition of the PBA-HKUST-1 composites is not given. What is the ratio between the two?

Response: We thank Reviewer 2 for the thoughtful suggestions. The actual composition ratio of the PBA-HKUST-1 has been measured by inductively coupled plasma-optical emission spectrometry (ICP-OES), showing a Cu/Fe molar ratio of 5.5:1.0. According to the molecular formulas of CuFe PBA ($Cu_3[Fe(CN)_6]_2$) and HKUST-1 ($C_{18}H_{12}Cu_3O_{15}$), the mass ratio of CuFe PBA and HKUST-1 in PBA-HKUST-1 is calculated to be ~2.7:7.3.

Corresponding discussions have been added in Page 7 in the revised manuscript as follows:

"In addition, the inductively coupled plasma-optical emission spectrometry (ICP-OES) shows a Cu/Fe molar ratio of 5.5:1.0. According to the molecular formulas of CuFe PBA ($Cu_3[Fe(CN)_6]_2$) and HKUST-1 ($C_{18}H_{12}Cu_3O_{15}$), the mass ratio of CuFe PBA and HKUST-1 in ONSA-HS-1 is calculated to be ~2.7:7.3."

Reviewer 3

General comments: In this manuscript, the authors report the synthesis of MOFs with an interesting house-of-cards architecture by orthogonal assembly of single crystal guest MOF nanosheets on a cubic host MOF. The structures of house-of-cards MOFs can be controlled from core-shell to hollow architectures with excellent catalytic properties. The materials have

been well characterized and the formation mechanism has also been analyzed in detail. Compared to reported MOF nanosheet assemblies, this work realizes a 3D ordered arrangement of MOF nanosheets, which is interesting. This manuscript is also well constructed and written. I recommend acceptance of this work after a few improvements.

Response: We thank Reviewer 3 for the positive comments.

Comment 1. Low-magnification SEM images of HoC-HS-2 and HoC-HS-3 should be provided.

Response: We thank Reviewer 3 for the useful comment. Low-magnification SEM images of ONSA-HS-2 and ONSA-HS-3 have been provided in the supporting information as below.

Fig. S14. Low-magnification SEM images of (A) ONSA-HS-2 and (B) ONSA-HS-3.

Comment 2. A scheme for illustrating the synthesis process should be added in SI.

Response: We appreciate Reviewer 3's constructive suggestions. A scheme for illustrating the synthesis process has been added in supplementary information as below.

Fig. S1. A scheme of the synthesis process for the ONSA heterostructures.

Comment 3. Except for $1/3\{220\}$, another group of diffraction points should also be indexed.

Response: Following Reviewer 3's useful suggestions, another group of diffraction points has been respectively indexed in Figs. 2B and C in the revised manuscript as below.

Fig. 2. Growth orientation of nanosheets in ONSA-HS-1. TEM images (A, C, E) and corresponding SAED patterns (B, D, F) of detached CuFe PBA nanosheet with Type I structure (A, B), Type II structure (C, D) and CuFe PBA nanosheets area in ONSA-HS-1 (E, F), respectively.

Comment 4. More TEM images of Type I and II nanosheets in other regions are recommended to be provided.

Response: We thank Reviewer 3 for this thoughtful comment. More TEM images of Type I and II nanosheets in other regions have been provided as fig. S10 in revised supplementary information as below.

Fig. S10. TEM images of detached CuFe PBA nanosheet with Type I structure (A, C), Type II structure (B, C).

Comment 5. The meanings of green, blue and red dots should be clarified in the figure caption.

Response: We acknowledge Reviewer 3's helpful comments. The green, red and blue dots in Fig. 3G represent the lattice points of (001) plane of HKUST-1, lattice points of (110) plane and ($1\bar{1}2$) of CuFe PBA, respectively.

Corresponding discussions have been added in Page 12 in the revised manuscript as follows:

"(G) Illustration of the epitaxy relationship between CuFe PBA and HKUST-1. The green, red and blue dots represent the lattice points of (001) plane of HKUST-1, (110) and ($1\bar{1}2$) planes of CuFe PBA, respectively."

Comment 6. The legends of e and f in Figure 5 are inconsistent.

Response: We are grateful Reviewer 3 for pointing out our mistakes. In the revised manuscript, another catalytic model (OER) has been used to replace the previous one. We have carefully checked the legends of each new figures. The newly collected results have been added as Fig.6 in the revised manuscript as below.

Fig. 6. OER performance of H-ONSA. (a) Structural advantages of H-ONSA for OER; (b) LSV curves, (c) overpotentials required for current density of 10 mA cm⁻², (d) Tafel slopes from LSV curves; (e) CV current density versus scan rate of different catalysts; the linear slope is equivalent to the double-layer capacitance (Cdl); (f) EIS spectra recorded at 1.479 V versus RHE; (g) LSV curves before and after successive CV scanning; (h) long-term stability of H-ONSA and RuO₂.

Figure 7. Optimized structures of the reaction intermediates on the CuFe PBA (111) surface (A) and (001) surface (B). (C) Free energy reaction profiles of the OER process at 1.23 V vs SHE.

Fig. S22. (A) TEM images and (B) XRD pattern of collapsed CuFe PBA nanosheets.

Fig. S23. (a, b) SEM images, (c) TEM image and corresponding (d) SAED pattern of CuFe PBA nanosheets-001.

Fig. S24. XRD pattern of CuFe PBA nanosheets-001.

Fig. S25. (A) SEM and (B) TEM images, and (C) XRD pattern of H-ONSA after cycling.

Fig. S26. CV curves between 1.35 and 1.45 V vs. RHE for (A) Nanocrystals, (B) Nanosheets, (C) H-ONSA and (D) CuFe PBA nanosheets-001.

Corresponding descriptions have been added in Page 17 in the revised manuscript as follows:

"Electrocatalytic OER performance

Oxygen evolution reaction (OER) plays important roles in various oxygen-involved energy conversion and storage technologies, where the development of effective electrocatalysts is the key (43, 44). Herein, the OER performances of H-ONSA were evaluated in 1.0 M KOH electrolyte with CuFe PBA nanocrystals, peeled CuFe PBA nanosheets (denoted as nanosheets-111, fig. S22) and commercial RuO₂ catalyst. In addition, to explore the effect of facet exposure, (001) dominated CuFe PBA nanosheets (denoted as nanosheets-001, fig. S23 and 24) were also synthesized. Fig. 6B shows the linear sweep voltammetry (LSV) curves of all samples. The H-ONSA requires lowest overpotential of 258 mV to reach the current density of 10 mA cm⁻², superior to those of CuFe PBA nanocrystals (430 mV), nanosheets-001 (350 mV), nanosheets-111 (301 mV) and even commercial RuO₂ (258 mV), indicating the highest OER activity (Fig. 6C). In addition, the OER kinetics of four catalysts were examined by Tafel plots (Fig. 6D). The H-ONSA exhibits a lower Tafel slope of 52.1 mV dec⁻¹ than nanocrystals (115.7 mV dec⁻¹), nanosheets-001 (96.0 mV dec⁻¹), nanosheets-111 (94.3 mV dec⁻¹) and RuO₂ (67.5 mV dec⁻¹), showing accelerated OER kinetics. Notably, the remarkable OER performances of H-ONSA is comparable to or even higher than most reported MOFs or other transition metal-based OER catalysts (Table S2).

Apart from the activity, stability is also crucial for OER electrocatalysts. Successive CV with a

scan rate of 50 mV s^{-1} was carried out. After 500 cycles, the obtained LSV curve of H-ONSA is similar to the initial one with slight increase of the overpotential (Fig. 6G), displaying a reasonable stability. The long-term durability of H-ONSA was further assessed by the chronoamperometric measurements (Fig. 6H). After 24 h, only about 7.4 % of current decay is observed in H-ONSA, significantly smaller than that of RuO_2 (high up to 44.5 %), suggesting the excellent OER stability. The sample after cycling test was also characterized by TEM, SEM and XRD, showing well-preserved morphology and crystalline structure (fig. S25).

To understand the superior OER performance of H-ONSA, the electrochemical surface area (ECSA) is further studied by calculating double-layer capacitances (C_{dl}) (45). The CV curves of different samples at different scan rates are presented in fig. S26. As expected, the H-ONSA affords more available active sites with a C_{dl} value of 34.8 mF cm^{-2} (Fig. 6E), which is ~ 9.9 , 3.5 and 3.1 times higher than that of nanocrystal (3.5 mF cm^{-2}), nanosheets-001 surface (9.97 mF cm^{-2}) and nanosheets-111 (11.3 mF cm^{-2}). Moreover, electrochemical impedance spectroscopy (EIS) was used to investigate the charge transport kinetics. The EIS spectra (Fig. 6F) show the lowest charge transfer resistance (R_{ct}) of H-ONSA among all samples, revealing the facilitated charge transfer.

To gain further insight into the facet exposure and OER activity relationship, DFT calculations were carried out to simulate the OER processes over (111) and (001) surfaces of CuFe PBA. The optimized structures of the OER intermediates on (111) and (001) surfaces are shown in Fig. 7, A and B, respectively. The corresponding free energy reaction profiles at $U = 1.23 \text{ V vs. SHE}$ are shown in Fig. 7C. In line with the experimental observation that CuFe PBA nanosheets with a (111) dominant surface exhibit enhanced OER performance, the accumulated free energy cost of OOH_{ad} generation, a rate-determining step during OER process (46,47), on the (111) surface is 1.10 eV, which is 0.26 eV lower than that on the (001) surface. The calculation results indicate that the superior OER activity of (111) surface can be attributed to two aspects. Firstly, the Fe atoms on the (111) surface are unsaturated and five-coordinated (as indicated by the red arrows in Fig. 7A), whereas on the (001) surface, they are saturated and six-coordinated (red arrows in Fig. 7B). As a result, the water molecules can directly adsorb onto the Fe sites on the (111) surface, while the replacement of -CN group is necessary for the water adsorption on the (001) surface. Secondly, on the (111) surface, the nearby -CN groups can stabilize the hydrogen-

containing intermediate species adsorbed on the surface via the formation of hydrogen bonds. Particularly, the formation of a strong hydrogen bond (with a bond length of 1.39 Å) leads to a reduced energy cost of the rate-determining $*O \rightarrow *OOH$ step on the (111) surface (0.40 eV), which is 0.15 eV lower than the energy cost on (100) surface (0.55 eV) in the absence of hydrogen bond formation.

Based on the above results, the enhanced OER performance of H-ONSA should be ascribed to the following structural superiorities (Fig. 6A). (1) The (111) dominated nanosheets offer unsaturated active sites and stabilize hydrogen-containing intermediate species, favoring the OER process. (2) The orthogonal arrangement of free-standing nanosheets reduces the likelihood of restacking of nanosheets, providing abundant surface active sites and thus improving the catalytic activity. (3) The created vertical and penetrating pore channels (48, 49), acting like a "highway", facilitates the diffusion of electrolytes and O₂ molecules. (4) The hollow structure is also conducive for the active site utilization and mass transfer (50, 51), enhancing the electrocatalytic properties."

Comment 7. The performance of 4-NP reduction in the absence of catalyst should be added.

Response: We acknowledge Reviewer 3's suggestions. In the revised manuscript, another catalytic model (OER) has been used to replace the previous one (see our response to your comment 6).

Comment 8. Several important references of MOF nanosheet assemblies should be cited (e.g. 10.1002/smt.201800068, 10.1021/jacs.0c02272).

Response: We thank Reviewer 3 for providing the highly relevant and important references, which have cited as Ref.14, 15 in revised manuscript.

"14. X. Wang et al., *Small Methods* **2**, 1800068 (2018).

15. T. Wang et al., *J. Am. Chem. Soc.* **142**, 20, 9408-9414 (2020)."

Reviewer 4

General comments: In their manuscript the authors describe the fabrication of hetero-aggregates, where cubic MOF (HKUST-I) particles are used as a substrate to grow 2D sheets of a Prussian-Blue analogue (PBA). The synthesis process yields "walls" of PBA nanosheets perpendicular to the substrate and orthogonal to each other.

The characterization of the constructs synthesized using the approach proposed by the authors appears to be adequate, it involves XRD, TEM, and BET measurements. All together, while I find the results interesting, I do not see that the properties of these constructs reaches the level of scientific excitement needed to warrant a publication in Nature Communications.

Response: We thank Reviewer 4 for the positive comments, stating that "The characterization of the constructs synthesized using the approach proposed by the authors appears to be adequate, it involves XRD, TEM, and BET measurements, and I find the results interesting". To address your concern about the properties of the constructs, we have conducted alkaline oxygen evolution reaction (OER, a core reaction in various energy conversion and storage technologies) as a new application (see details below). Based on the new results, we believe that the scientific significance and novelty of our work has been greatly improved from the property perspective as follows: (1) the H-ONSA exhibited a low overpotential of 245 mV at current density of 10 mA cm⁻², low Tafel slope of 52.1 mV dec⁻¹ and excellent stability (current decay of 7.4% after 24 h). The remarkable OER performances even exceeded commercial noble metal (RuO₂) and were also higher than most reported MOFs or other transition metal-based OER catalysts; (2) To our knowledge, this is the first report of a new composition (CuFe PBA) and a new structure (orthogonal nanosheet array) in the application of electrocatalysis. The superiority of a naturally nonpreferred (111) facet of CuFe-PBA in electrocatalysis is also unveiled for the first time. Compared to reported CuFe PBA nanosheets with (100) exposed facets, those with (111) exposed facets offer abundant unsaturated binding sites and stabilize the intermediate species via hydrogen bonding in oxygen evolution reaction (OER) as evidenced by the theoretical calculation results. Together with the promoted active site exposure and mass transfer by macroporous array of nanosheets, the OER performance of H-ONSA is enhanced than conventional CuFe PBA nanosheets. Together with the facilitated mass diffusion and active site exposure by the orthogonal arrangement of nanosheets that create large pore channels, the OER

performance of H-NOSA is significantly reinforced. To our knowledge, this is the first report of a new composition (CuFe PBA) and a new structure (orthogonal nanosheet array) in the application of electrocatalysis. The superiority of a naturally nonpreferred (111) facet of CuFe-PBA in electrocatalysis is also unveiled for the first time.

The newly collected OER results have been shown below.

Fig. 6. OER performance of H-ONSA. (a) Structural advantages of H-ONSA for OER; (b) LSV curves, (c) overpotentials required for current density of 10 mA cm⁻², (d) Tafel slopes from LSV curves; (e) CV current density versus scan rate of different catalysts; the linear slope is equivalent to the double-layer capacitance (Cdl); (f) EIS spectra recorded at 1.479 V versus RHE; (g) LSV curves before and after successive CV scanning; (h) long-term stability of H-ONSA and RuO₂.

Figure 7. Optimized structures of the reaction intermediates on the CuFe PBA (111) surface (A) and (001) surface (B). (C) Free energy reaction profiles of the OER process at 1.23 V vs SHE.

Fig. S22. (A) TEM images and (B) XRD pattern of collapsed CuFe PBA nanosheets.

Fig. S23. (A, B) SEM images, (C) TEM image and corresponding (D) SAED pattern of CuFe PBA nanosheets-001.

Fig. S24. XRD pattern of CuFe PBA nanosheets-001.

Fig. S25. (A) SEM and (B) TEM images, and (C) XRD pattern of H-ONSA after cycling.

Fig. S26. CV curves between 1.35 and 1.45 V vs. RHE for (A) Nanocrystals, (B) Nanosheets, (C) H-ONSA and (D) CuFe PBA nanosheets-001.

Corresponding descriptions have been added in Page 17 in the revised manuscript as follows:

"Electrocatalytic OER performance

Oxygen evolution reaction (OER) plays important roles in various oxygen-involved energy conversion and storage technologies, where the development of effective electrocatalysts is the key (43, 44). Herein, the OER performances of H-ONSA were evaluated in 1.0 M KOH electrolyte with CuFe PBA nanocrystals, peeled CuFe PBA nanosheets (denoted as nanosheets-111, fig. S22) and commercial RuO₂ catalyst. In addition, to explore the effect of facet exposure, (001) dominated CuFe PBA nanosheets (denoted as nanosheets-001, fig. S23 and 24) were also synthesized. Fig. 6B shows the linear sweep voltammetry (LSV) curves of all samples. The H-ONSA requires lowest overpotential of 258 mV to reach the current density of 10 mA cm⁻², superior to those of CuFe PBA nanocrystals (430 mV), nanosheets-001 (350 mV), nanosheets-111 (301 mV) and even commercial RuO₂ (258 mV), indicating the highest OER activity (Fig. 6C). In addition, the OER kinetics of four catalysts were examined by Tafel plots (Fig. 6D). The H-ONSA exhibits a lower Tafel slope of 52.1 mV dec⁻¹ than nanocrystals (115.7 mV dec⁻¹), nanosheets-001 (96.0 mV dec⁻¹), nanosheets-111 (94.3 mV dec⁻¹) and RuO₂ (67.5 mV dec⁻¹), showing accelerated OER kinetics. Notably, the remarkable OER performances of H-ONSA is comparable to or even higher than most reported MOFs or other transition metal-based OER catalysts (Table S2).

Apart from the activity, stability is also crucial for OER electrocatalysts. Successive CV with a

scan rate of 50 mV s^{-1} was carried out. After 500 cycles, the obtained LSV curve of H-ONSA is similar to the initial one with slight increase of the overpotential (Fig. 6G), displaying a reasonable stability. The long-term durability of H-ONSA was further assessed by the chronoamperometric measurements (Fig. 6H). After 24 h, only about 7.4 % of current decay is observed in H-ONSA, significantly smaller than that of RuO_2 (high up to 44.5 %), suggesting the excellent OER stability. The sample after cycling test was also characterized by TEM, SEM and XRD, showing well-preserved morphology and crystalline structure (fig. S25).

To understand the superior OER performance of H-ONSA, the electrochemical surface area (ECSA) is further studied by calculating double-layer capacitances (C_{dl}) (45). The CV curves of different samples at different scan rates are presented in fig. S26. As expected, the H-ONSA affords more available active sites with a C_{dl} value of 34.8 mF cm^{-2} (Fig. 6E), which is ~ 9.9 , 3.5 and 3.1 times higher than that of nanocrystal (3.5 mF cm^{-2}), nanosheets-001 surface (9.97 mF cm^{-2}) and nanosheets-111 (11.3 mF cm^{-2}). Moreover, electrochemical impedance spectroscopy (EIS) was used to investigate the charge transport kinetics. The EIS spectra (Fig. 6F) show the lowest charge transfer resistance (R_{ct}) of H-ONSA among all samples, revealing the facilitated charge transfer.

To gain further insight into the facet exposure and OER activity relationship, DFT calculations were carried out to simulate the OER processes over (111) and (001) surfaces of CuFe PBA. The optimized structures of the OER intermediates on (111) and (001) surfaces are shown in Fig. 7, A and B, respectively. The corresponding free energy reaction profiles at $U = 1.23 \text{ V vs. SHE}$ are shown in Fig. 7C. In line with the experimental observation that CuFe PBA nanosheets with a (111) dominant surface exhibit enhanced OER performance, the accumulated free energy cost of OOH_{ad} generation, a rate-determining step during OER process (46,47), on the (111) surface is 1.10 eV, which is 0.26 eV lower than that on the (001) surface. The calculation results indicate that the superior OER activity of (111) surface can be attributed to two aspects. Firstly, the Fe atoms on the (111) surface are unsaturated and five-coordinated (as indicated by the red arrows in Fig. 7A), whereas on the (001) surface, they are saturated and six-coordinated (red arrows in Fig. 7B). As a result, the water molecules can directly adsorb onto the Fe sites on the (111) surface, while the replacement of -CN group is necessary for the water adsorption on the (001) surface. Secondly, on the (111) surface, the nearby -CN groups can stabilize the hydrogen-

containing intermediate species adsorbed on the surface via the formation of hydrogen bonds. Particularly, the formation of a strong hydrogen bond (with a bond length of 1.39 Å) leads to a reduced energy cost of the rate-determining $*O \rightarrow *OOH$ step on the (111) surface (0.40 eV), which is 0.15 eV lower than the energy cost on (100) surface (0.55 eV) in the absence of hydrogen bond formation.

Based on the above results, the enhanced OER performance of H-ONSA should be ascribed to the following structural superiorities (Fig. 6A). (1) The (111) dominated nanosheets offer unsaturated active sites and stabilize hydrogen-containing intermediate species, favoring the OER process. (2) The orthogonal arrangement of free-standing nanosheets reduces the likelihood of restacking of nanosheets, providing abundant surface active sites and thus improving the catalytic activity. (3) The created vertical and penetrating pore channels (48, 49), acting like a "highway", facilitates the diffusion of electrolytes and O_2 molecules. (4) The hollow structure is also conducive for the active site utilization and mass transfer (50, 51), enhancing the electrocatalytic properties."

Comment 1. "House of Cards", the term used by the authors, is not appropriate. A house of cards consists of cards not only standing perpendicular on a substrate (table) but also contains cards with an orientation parallel to the substrate. (<https://www.youtube.com/watch?v=SEBBj2BIBm8>).

Response: We thank Reviewer 4 for the critical comment. According to your suggestion, the term "house of cards" has been replaced by "orthogonal nanosheets with single-crystalline arrays (ONSA)" to describe the ordered arrangement of CuFe PBA nanosheets on HKUST-1.

Comment 2. The notion "Wulff's point", a term used by the authors, is unknown to me.

Response: Wulff's theorem is a theory to determine the equilibrium shape of a crystal of fixed volume inside a separate phase. Specifically, the minimum surface energy for a given volume of a crystal will be achieved if the distances of its faces from one fixed point are proportional to their capillary constants, where the point is called as "Wulff's point". On this basis, the

Wulff's theory can be used to understand the growth of crystals with specific morphologies on the substrates, for example epitaxial nickel oxide on the strontium titanate (J. Phys. Chem. C 2021, 125, 23, 12827-12836), and semi-hexagonal CuFe PBA nanosheet on HKUST-1 in our work.

Reviewer comments, second round

Reviewer #1 (Remarks to the Author):

Although this paper has undergone intensive revisions and added more experimental results on the electrocatalytic OER performance, it seems insufficient to make remarkable advances. In order to compare the electrocatalytic OER performance as an additional application of H-ONSA, it is necessary to include a comparison with other hollow shells of Prussian blue analogues that do not have a specific array. The electrocatalytic OER performance of H-ONSA mentioned in this paper appears to show inferior performance compared to other hollow shell structures (Adv. Func. Mater., 2021, 31, 210685, J. Am. Chem. Soc., 2014, 136, 19, 7077–7084, Adv. Mater., 2017, 29, 1703870). Additionally, some additional points are listed below:

Benchmarking the OER activity of H-ONSA with that of previously reported Prussian Blue-derived OER catalysts should be conducted.

Coordination compounds such as Prussian Blue- or MOF-derived catalysts typically undergo structural and/or compositional transformations during/after electrocatalytic reactions. This issue should be addressed using post-mortem analyses.

The intrinsic OER activity of the compared catalysts should be assessed by turnover frequency. In this work, this can be achieved by dividing the current density by ECSAs.

Reviewer #3 (Remarks to the Author):

Authors have carefully addressed all the concerns from this reviewer. I would like to recommend its publication in current form.

Reviewer #4 (Remarks to the Author):

The authors have submitted the manuscript to an extensive revision in following the suggestions of the referees. During this process, the quality of the presentation has been improved substantially. All criticism was carefully considered and I find the answers mostly satisfactory.

I am happy to recommend publication of the revised version of the manuscript for publication in Nature Communications.

Point-to-point response and revisions

Reviewer 1

General comments: Although this paper has undergone intensive revisions and added more experimental results on the electrocatalytic OER performance, it seems insufficient to make remarkable advances. In order to compare the electrocatalytic OER performance as an additional application of H-ONSA, it is necessary to include a comparison with other hollow shells of Prussian blue analogues that do not have a specific array. The electrocatalytic OER performance of H-ONSA mentioned in this paper appears to show inferior performance compared to other hollow shell structures (Adv. Func. Mater., 2021, 31, 210685, J. Am. Chem. Soc., 2014, 136, 19, 7077-7084, Adv. Mater., 2017, 29, 1703870).

Response: We thank Reviewer 1 for the constructive comments helping us to improve the quality of our work. Following your instruction, a comprehensive comparison of the OER activities with previously reported PBA-based catalysts including the other hollow shells of Prussian blue analogues has been provided in Table S2 (see also below). The OER activity of our designed H-ONSA is higher or comparable to most reported PBA-based materials.

Besides, we have also made a careful comparison with the three references provided by Reviewer 1, which have been cited as Ref. 29-31 in the revised manuscript. In the three references, Ni-Fe mixed diselenide nanocages (Adv. Mater., 2017, 29, 1703870), α -Ni(OH)₂ (J. Am. Chem. Soc., 2014, 136, 7077) and NiCo@A-NiCo-PBA (Adv. Func. Mater., 2021, 31, 2106835) were used as OER electrocatalysts, which required an overpotential of 240, 331 and 276 mV at a current density of 10 mA cm⁻², respectively. For our H-ONSA, the overpotential of 241 mV is 36 and 91 mV lower than that of Ni-Fe mixed diselenide nanocages and α -Ni(OH)₂, and only 1 mV higher than that of NiCo@A-NiCo-PBA. The OER performance of H-ONSA is higher than the electrocatalysts in JACS and AFM papers, and very close to the electrocatalyst in AM paper.

To accurately describe the OER performance of our material, the following statement has been made in Page 18 in the revised manuscript:

"The OER performance of H-ONSA is comparable to most reported PBA-derived electrocatalysts

or other MOFs and transition metal-based materials (Table S2)."

Table S2. Comparison of OER activity of H-ONSA and recently reported MOF-based catalysts in KOH solution.

Samples	Overpotential/mV (at 10 mA cm ⁻²)	Tafel/mV dec ⁻¹	Ref.
H-ONSA	241	52.1	This work
Co-MOF	320	142	17
NNU-23	365	81.8	18
Ni ₂ Fe ₁ Sq-zbr-MOF	230	37.7	19
Ligand mixed MOF-Fe	288	39	20
NiFe-MOFs	258	19	21
CoFe-PBA@NF-24	256	54	22
Ar-U-CoFe PBA	305	36.1	23
CoMoS	282	58	24
NiFe-DASC	310	45	25
meso-Fe-MoS ₂ /CoMo ₂ S ₄	290	65	26
Ni-Co-Fe LDH nanosheet	288	92	27
Co _{0.25} Fe _{0.75} O ₄	350	50	28
α-Ni(OH) ₂	331	42	29
Ni-Fe mixed diselenide nanocages	240	24	30
NiCo@A-NiCo-PBA-AA	276	79.1	31
CoFe PBA/CoS ₂ -12 CNBs	265	59.2	32
O-GQD-NiFe PBA	259	52.5	33
NiCoFe-P-NP @ NiCoFe-PBA	223	78	34
PBA-5	271	53.7	35
H-SCP-350	291	38	36
CF-PBA-400	254	51	37
Co PBA-O/Ce-2	240	137	38
O-CNT/NiFe 1:18	279	42.8	39
NiFe-PBA	258	46	40
FeCoS _x -PBA	266	33	41

Comment 1. Benchmarking the OER activity of H-ONSA with that of previously reported Prussian Blue-derived OER catalysts should be conducted.

Response: We appreciate Reviewer 1's useful suggestion. A detailed comparison of OER activities with previously reported PBA-based catalysts has been added in the revised manuscript (see details in the response to your general comments).

Comment 2. Coordination compounds such as Prussian Blue- or MOF-derived catalysts typically undergo structural and/or compositional transformations during/after electrocatalytic reactions. This issue should be addressed using post-mortem analyses.

Response: We are grateful for Reviewer 1's valuable comments. XPS measurement was conducted to study the changes of the surface structure and chemical states of H-ONSA before and after OER test. The results (see below, Figure S26) indicate a surface reconstruction of the metal ions into metal (oxy)hydroxides during OER process, in agreement with the reported Prussian Blue- or MOF-derived catalysts (Adv. Mater. 2023, 35, 2208904; Nat. Energy 2020, 5, 881).

Fig. S26. High-resolution XPS spectra of Fe 2p (A), Cu 2p (B) and O 1s (C) of H-ONSA and H-ONSA-A.

Corresponding descriptions have been added in Page 19 in the revised manuscript as follows:

“The changes of the surface structure and chemical states of H-ONSA before and after OER test were further studied by XPS (the sample after use was denoted as H-ONSA-A). In the Fe 2p spectrum of H-ONSA (fig. S26A), the peaks at 708.2 and 721.0 eV are assigned to Fe²⁺. In addition, two peaks at 710.0 and 723.7 eV are attributed to the 2p_{3/2} and 2p_{1/2} states of Fe³⁺, with two satellite peaks at 787.9 and 804.1 eV. The Fe³⁺/Fe²⁺ ratio was determined to be 1.11. The Cu 2p spectrum of H-ONSA (fig. S26B) shows four main peaks, attributed to 2p_{1/2} and 2p_{3/2} orbitals of Cu²⁺ (955.8 and 935.8 eV) and Cu⁺ (952.7 and 933.0 eV), and satellite peaks with Cu²⁺/Cu⁺ ratio of 0.71. Compared to the fresh H-ONSA, H-ONSA-A exhibited similar Fe and Cu species with higher Fe³⁺/Fe²⁺ and Cu²⁺/Cu⁺ ratios (1.44 and 2.03, respectively). For O 1s spectrum of H-ONSA (fig. S26C), the peak at 532.5 eV is assigned to the -OH group. The observation of a new peak of -OOH group at 535.4 eV for H-ONSA-A indicates the generation of metal (oxy)hydroxides during OER process, consistent with the literature reports of MOF-based OER electrocatalysts (45, 46).”

Comment 3. The intrinsic OER activity of the compared catalysts should be assessed by turnover frequency. In this work, this can be achieved by dividing the current density by ECSAs.

Response: We acknowledge Reviewer 1’s important comment. To assess the intrinsic OER activity of the catalysts, the turnover frequency (TOF) values including both lower-bound TOF (TOF_{lb}) and upper-bound TOF (TOF_{ub}) have been calculated by normalizing the current density with ECSA values and mass loading of catalysts at an overpotential of 240 mV (see details below). Specifically, all of the metal sites in the catalyst are assumed to be the active sites for calculating TOF_{lb} and the value is $1.84 \times 10^{-4} \text{ s}^{-1}$. Besides, the TOF_{ub} is determined to be 352.59 s^{-1} by using the surface concentration of active metal sites.

The following descriptions have been added in the revised manuscript.

"Calculation of turnover frequencies²

Turnover frequency (TOF) is an important quantitative metric for electrocatalysts and can be calculated by

$$\text{TOF} = j \times A / (4 \times F \times n),$$

where j is the current density normalized by ECSA values and mass loading of catalyst at the given overpotential, A is the area of the electrode, F is the Faraday constant (96485 C mol^{-1}), and n is the number of moles of active sites of the metal in the catalysts, which can be calculated by two methods.

- 1) **Lower bound TOF (TOF_{lb}):** The TOF was calculated by assuming that all the metal sites in H-ONSA are active to catalysis according to the following equation:

$$n = \% \text{ mass of metal in catalyst} \times \text{mass of catalyst} / \text{molecular weight of metal}$$

- 2) **Upper bound TOF (TOF_{ub}):**³ The TOF was calculated by assuming that only the surface metal sites in the compounds are active to catalysis by using following method^{4,5}:

The number of surface metal atoms in the H-ONSA was calculated by assuming that the (111) crystal face is exposed in all cases. The (111) facet in a unit cell gives 6 Fe atoms and 6 Cu atoms with an area of $\sqrt{3}a^2$, where a (10.10 \AA) is the unit cell edge length of CuFe PBA. The density of surface metal atoms is thus calculated to be 6.79189×10^{18} surface metal atoms/ m^2 . Further according to the mass loading for OER test (0.05 mg) and surface area ($113.4 \text{ m}^2/\text{g}$, Fig. S28) of H-ONSA, the number of total surface metal atoms was determined to be 3.85201×10^{16} ."

"To assess the intrinsic activity of H-ONSA, the turnover frequency (TOF, see the calculation details in supplementary materials) including lower-bound TOF (TOF_{lb}) and upper-bound TOF (TOF_{ub}) was calculated to be 1.84×10^{-4} and 352.59 s^{-1} at an overpotential of 240 mV , respectively."

Fig. S28.

Nitrogen adsorption-desorption isotherms of H-ONSA. The BET specific surface area is $113.4 \text{ m}^2/\text{g}$.

Reviewer 3

General comments: Authors have carefully addressed all the concerns from this reviewer. I would

like to recommend its publication in current form.

Response: We thank Reviewer 3 for the positive comments.

Reviewer 4

General comments: The authors have submitted the manuscript to an extensive revision in following the suggestions of the referees. During this process, the quality of the presentation has been improved substantially. All criticism was carefully considered and I find the answers mostly satisfactory.

I am happy to recommend publication of the revised version of the manuscript for publication in Nature Communications.

Response: We appreciate Reviewer 4's positive comments.

Reviewer comments, third round

Reviewer #1 (Remarks to the Author):

The authors have carefully addressed all the concerns. I would like to recommend its publication as it is.